# Code2Video: A Code-centric Paradigm for Educational Video Creation

Yanzhe Chen [* 1]   Kevin Qinghong Lin [* 2]   Mike Zheng Shou [1]

https://showlab.github.io/Code2Video/

## Abstract

While recent generative models can synthesize videos in pixel space, they often fail to produce educational videos with precise structures, domain knowledge, and coherent transitions. We argue that this setting is better served by operating in a renderable environment that is explicitly controlled by code. We propose **Code2Video**, a code-centric agent framework that generates educational videos by writing executable Python programs. Code2Video includes three agents: a *Planner* that converts lecture content into a temporal storyboard, a *Coder* that turns the storyboard into runnable code with scope-guided auto-fix, and a *Critic* that refines layout using a VLM guided by *visual anchor prompting*, *i.e.,* mappings from target visual outcomes to code edits. For evaluation, we build **MMMC**, a benchmark of professionally produced, discipline-specific educational videos. We assess Code2Video using aesthetic scores (VLM-as-a-Judge), code efficiency, and **TeachQuiz**, an end-to-end metric that measures how well an *unlearned* VLM can recover knowledge after watching generated videos. Code2Video improves performance by 40% over direct code generation and produces videos comparable to human-crafted tutorials. The code and datasets are available at https://github.com/showlab/Code2Video.

## 1. Introduction

> *"If you want to master something, teach it."* –
> Richard Feynman

Recent progress in video creation has largely happened in *pixel* space. End-to-end models, including diffusion (Ho et al., 2022a; Weng et al., 2024b) and autoregressive architectures (Weng et al., 2024a; Yuan et al., 2025), can synthesize visually compelling videos from text prompts (*i.e.,* **Text2Video**). They often achieve strong short-form fidelity. However, they remain brittle on long-form reasoning and complex multi-entity interactions (Li et al., 2024a). To address this, recent systems adopt multi-agent pipelines that decompose generation into subtasks, enabling iterative refinement and temporal structuring (Yuan et al., 2024b; Huang et al., 2024; Xie et al., 2024).

Educational videos designed to teach subject-specific knowledge face unique challenges. Unlike short-form entertainment, educational content must integrate deep domain expertise (Clark & Mayer, 2023), carefully designed animations or transitions, and step-by-step reasoning (Bao et al., 2009; Fencl, 2010) to support actual skill acquisition. This raises two fundamental challenges: **(i)** How to create high-quality educational videos that maintain both temporal coherence—concepts introduced, expanded, and reinforced in logical sequence—and spatial clarity—elements arranged legibly without occlusion; and **(ii)** How to evaluate educational videos beyond appearance, ensuring that they are educationally effective and semantically aligned with the intended learning topic. Existing video generation pipelines rarely satisfy these requirements, leaving a critical gap for agentic methods that unify temporal planning, spatial organization, and educational assessment.

We are motivated by the unique advantages of code for educational video creation. Unlike black-box models, code-centric pipelines are *scalable*, since new visualizations and external assets can be modularly integrated; *interpretable*, as every sequence, layout, and rendering decision is explicitly scripted and thus auditable; and *controllable*, enabling precise temporal sequencing and spatial organization through programmatic specification.

Building on this, we propose **Code2Video**, a code-centric agent framework that generates educational videos by writing executable Manim programs. Code2Video consists of three agents. The *Planner* organizes concepts, examples, and recaps into a lecture flow. The *Coder* turns the plan into runnable Manim code with consistent layout and timing. The *Critic* improves spatial organization with multimodal

---

[*]Equal contribution   [1]Show Lab, National University of Singapore [2]University of Oxford. Correspondence to: Mike Zheng Shou <mike.zheng.shou@gmail.com>.

*Proceedings of the 43rd International Conference on Machine Learning*, Seoul, South Korea. PMLR 306, 2026. Copyright 2026 by the author(s).

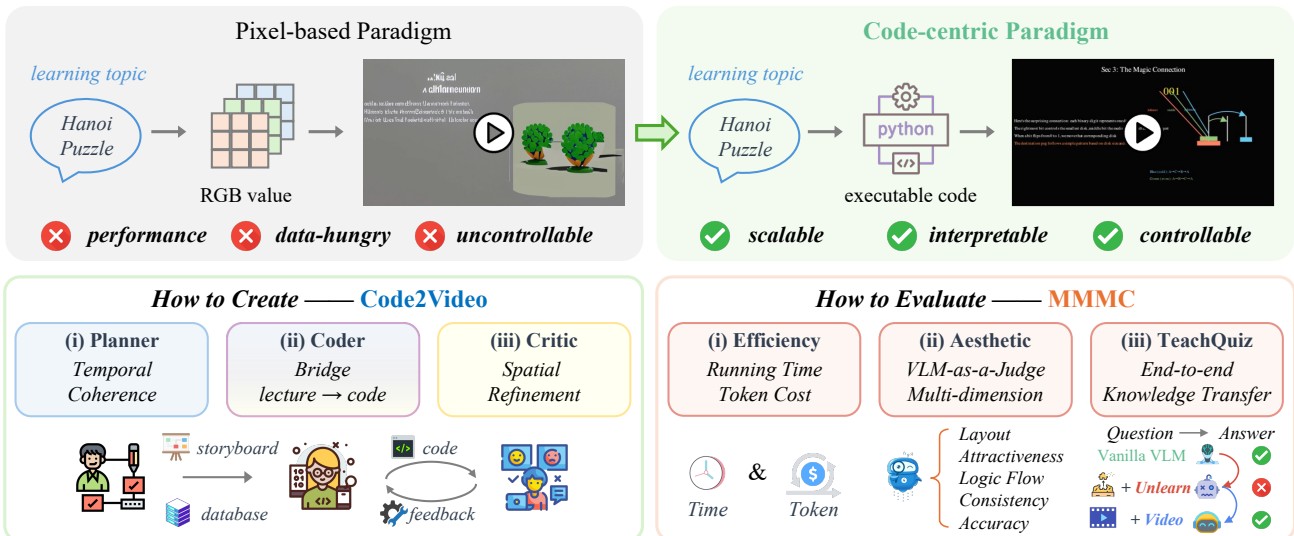

Figure 1. **The Code2Video framework and evaluation protocol.** We introduce a code-centric paradigm for generating educational videos, enabling precise controllability and interpretability compared to pixel-based methods. The framework uses collaborative agents to synthesize executable code, evaluated on our proposed **MMMC** benchmark.

feedback and *visual anchor points*. A visual anchor point specifies a target visual outcome and the corresponding code-level edits, linking *what we want to see* to *how to change the program*. This design makes the temporal and spatial structure explicit, while keeping the pipeline transparent and reproducible.

To evaluate this paradigm, we introduce **MMMC** (**M**assive **M**ulti-discipline **M**ultimodal **C**oding), a benchmark of educational videos designed to measure knowledge transfer. MMMC contains professionally produced, discipline-specific Manim tutorials across 13 areas (*e.g.,* topology, physics). We evaluate along three dimensions: (i) VLM-as-a-Judge scores for aesthetics and structure; (ii) code efficiency, including generation time and token usage; and (iii) **TeachQuiz**, an end-to-end knowledge-transfer metric. TeachQuiz first unlearns a target concept from a VLM, and then measures how well the generated video restores that concept. Our experiments reveal several key findings. Pixel-based models struggle with fine details and long-form coherence. A direct code-generation baseline improves TeachQuiz by 30%. Our full pipeline yields a stable 40% gain. In human studies aligned with TeachQuiz, our videos are preferred over strong baselines and are competitive with professional tutorials.

Our contributions are summarized as follows:

- **A New Paradigm for Video Creation.** We introduce a new code-centric paradigm for educational video creation, positioning executable code as the unifying medium for temporal sequencing and spatial organization.

- **Effective Designs for Visual Animation Agent.** We

propose a modular agent framework with three key components: (i) The *Planner* expands an external database for reference, enabling parallel yet consistent storyboard; (ii) The *Coder* ensures executable code via automatic debugging and scope-guided repair; (iii) The *Critic* refines spatial layout and clarity using visual anchor prompting.

- **A New Benchmark with Well-designed Evaluation Protocol.** We present MMMC, the first benchmark for code-centric educational video creation with multi-dimensional evaluation of efficiency, aesthetics, and end-to-end knowledge transfer.

## 2. Related Work

### 2.1. Video Generation

We review three lines of work. **(i) Pixel-space text-to-video synthesis.** Early methods extend diffusion models to the temporal domain using space–time UNets and latent 3D VAEs (Weng et al., 2024b; Ho et al., 2022b), achieving strong perceptual quality and longer durations (Yang et al., 2024; Li et al., 2024a; Xing et al., 2024). However, pixel-level synthesis offers limited control over exact layouts and symbolic alignment, which are central to educational videos. Recent efforts improve long-form consistency (Li et al., 2024b; Gu et al., 2025; Wang et al., 2024; Xie et al., 2025; Lu et al., 2024; Zhou et al., 2024), yet still struggle with board-like composition and stepwise exposition (Li et al., 2024a; Liu et al., 2024). **(ii) Multi-agent pipelines.** Multi-agent systems decompose complex goals into subtasks, coordinate tools, and iteratively refine output, which can improve reasoning and generation (Yuan et al., 2024b;

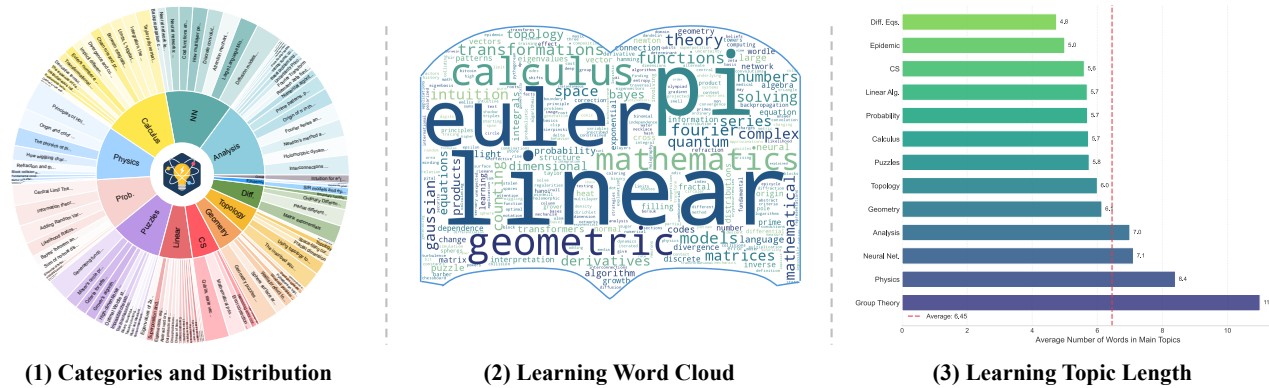

**(1) Categories and Distribution**  **(2) Learning Word Cloud**  **(3) Learning Topic Length**

*Figure 2.* **MMMC overview**. (1) Left: Distribution of 13 subject areas with exemplar learning topics; the ring width represents video duration. (3) Right: Average learning topic length subject per area. For a clearer version of the left figure, please refer to Fig. 7.

Hu et al., 2024; Xie et al., 2024; Shen et al., 2024). Their use in video creation is still relatively limited compared to other domains (Ku et al., 2025; Wu et al., 2024b). **(iii) Renderable environments and code-centric video creation.** Beyond pixel-space synthesis, videos can also be generated by controlling a renderable environment (e.g., engines or executable code), enabling deterministic replay and editable structure (Ku et al., 2025; Yin et al., 2026; He et al., 2024; Cai et al., 2024). We adopt this broader view of video creation as an input→output mapping to coherent videos, and focus on code-centric educational content. Executable code supports symbolic layout, temporally structured exposition, and reproducible rendering—hard to guarantee with pixel-level diffusion.

## 2.2. Coding Agents

Recent advances in LLM-based tool use demonstrate that agents can autonomously call APIs, retrieve information, and verify outputs. This capability enables robust task decomposition (Yao et al., 2023; Wang et al., 2025). By integrating code execution and tool invocation, representative methods extend language models beyond **text-only** reasoning, supporting complex workflows and project-level code generation (Patil et al., 2024; Liu et al., 2025; Gupta et al., 2024). Such developments demonstrate the potential of LLM agents to coordinate external retrieval, maintain memory across parallel processes, and incorporate feedback loops for iterative refinement (Li, 2025; Xu et al., 2025; Zhang et al., 2024). In parallel, research at the intersection of coding and visual reasoning shows that generating and executing code can yield structured perception and controllable rendering (Pang et al., 2025; Zhu et al., 2025; Lin et al., 2025). **Visual programming** and visual-to-code approaches leverage program synthesis for compositional reasoning and spatial arrangement, with benchmarks translating images or text into executable code for charts, plots, and graphical interfaces (Wu et al., 2024a; Zhao et al., 2025; Wei et al.,

2025; Yen et al., 2025). While these works bridge symbolic and visual domains, they largely focus on *static* figures or localized visual tasks (Xing et al., 2025; Wen et al., 2024; Ye et al., 2025; Jain et al., 2025). We advance this line by integrating code generation and visual synthesis for *dynamic* educational **video creation**.

## 3. MMMC Benchmark

### 3.1. Task Formulation

The task of code-centric educational video generation maps a learning query to executable *Manim* (Manim Community Dev, 2025) code whose rendering yields a tutorial video. The challenge lies in multi-step reasoning, precise temporal sequencing, and spatial coherence, where minor syntax errors can prevent successful rendering. We adopt *Manim* for its fine-grained control, symbolic expressivity, and demonstrated effectiveness in expert-produced instructional videos.

### 3.2. Data Curation and Statistics

We construct MMMC, a benchmark for code-driven educational video creation, under two criteria: (i) *educational relevance*—each learning topic is an established concept worth teaching; and (ii) *executable grounding*—each concept aligns with a high-quality Manim reference, ensuring practical realizability. We download videos from the 3Blue1Brown (3B1B) YouTube channel, known for its instructional impact and expert Manim craftsmanship. After filtering out non-instructional items, we curate 117 long-form videos spanning 13 subject areas, including *calculus*, *geometry*, *probability*, and *neural networks*. We segmented these into 339 sub-clips using timestamps, resulting in 456 total units. Using an LLM, we extracted concise learning topics (avg. 6.3 words) from the metadata, creating a clean mapping from videos to educational units (details in §A.1.5). On average, a full-length video lasts 1014 seconds (~16.9

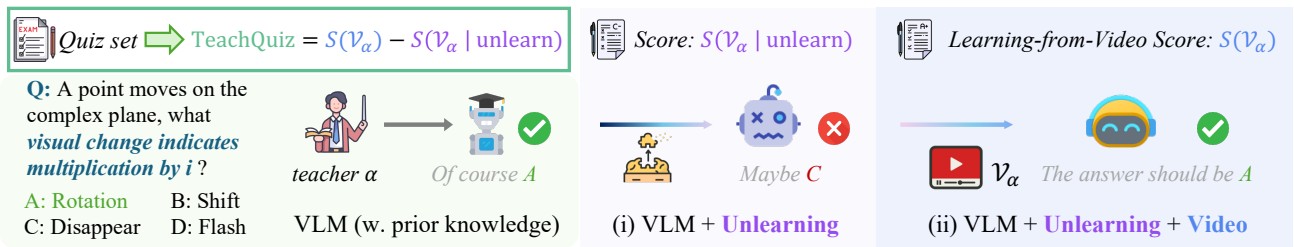

*Figure 3.* TeachQuiz: score gap between *Learning-from-Video* and *Unlearning* stages.

minutes), while a segmented clip spans 201 seconds (~3.35 minutes), thus balancing long-horizon reasoning with fine-grained supervision. Fig. 2 visualizes topical diversity with a hierarchical donut plot: the inner ring denotes 13 categories, and the outer ring shows individual topics, with the arc width proportional to the cumulative duration. This structure highlights the breadth of coverage and temporal richness of MMMC, establishing a challenging and representative benchmark for educational video generation.

### 3.3. Evaluation Metrics

Unlike conventional video generation, the value of educational videos lies less in visual fidelity and more in how effectively they convey knowledge. Since standard synthesis metrics are insufficient, we propose a multi-dimensional evaluation framework assessing **Aesthetics**, **Efficiency**, and critically, **Knowledge Transfer**.

**VLM-as-Judge (Aesthetics).** We use a structured VLM prompt $\mathcal{P}_{\text{aesth}}$ to score five axes on a 100-point scale: *Element Layout*, *Attractiveness*, *Logic Flow*, *Visual Consistency*, and *Accuracy & Depth*. These axes capture readability, coherence, and conceptual correctness.

**TeachQuiz (Knowledge Conveyance).** We introduce *TeachQuiz* to measure concept transfer from video, as illustrated in Fig. 3. For each concept $\mathcal{K}$, we build a quiz set $\mathcal{Q}(\mathcal{K}) = \{(q_i, y_i)\}_{i=1}^{N}$. Given a video $\mathcal{V}_\alpha$, let $S(\mathcal{V}_\alpha)$ denote the model accuracy after watching $\mathcal{V}_\alpha$:

$$S(\mathcal{V}_\alpha) = \frac{1}{N} \sum_{i=1}^{N} \mathbf{1}\big[\phi(q_i \mid \mathcal{V}_\alpha) = y_i\big]. \qquad (1)$$

**Why a controlled baseline is needed.** A key challenge is that a model's quiz accuracy depends on both its video understanding ability and its pre-existing knowledge. This becomes problematic with powerful closed-source VLMs, as *many quiz items can be answered correctly even without watching the video*, making raw accuracy an unreliable measure of a video's teaching quality. To address this with black-box models that cannot be fine-tuned, we use a training-free *in-context unlearning* step as a *control intervention*. Our goal is not to advance unlearning, but to

suppress prior knowledge of $\mathcal{K}$ enough to make post-video gains measurable and reproducible.

**(i) Unlearning.** We employ in-context unlearning (Pawelczyk et al., 2023; Takashiro et al., 2025) to establish a knowledge-depleted baseline. This approach operates on the principle that a model's output distribution can be guided via instructional prompts, effectively simulating "forgetting" within a black-box paradigm (Thaker et al., 2024; Geng et al., 2025; Pawelczyk et al., 2023). Our prompt $\mathcal{P}_{\text{unlearn}}$ instructs the model to suppress any pre-existing knowledge of concept $\mathcal{K}$-including definitions, formulas, and solution heuristics—and to default to responding with INSUFFICIENT EVIDENCE for related queries. This induces a significant accuracy drop on $\mathcal{Q}(\mathcal{K})$, creating a controlled pre-instruction state. The efficacy of this unlearning step is empirically validated in our experiments (§A.1.1 and §5.3).

**(ii) Learning-from-Video.** Expose the model to $\mathcal{V}$ under prompt $\mathcal{P}_{\text{learn}}$, testing whether the video itself enables recovery of the knowledge. We define the *TeachQuiz score* $\widetilde{S}$ as the improvement over the unlearned baseline:

$$\widetilde{S}(\mathcal{V}_\alpha) = S(\mathcal{V}_\alpha) - S(\mathcal{V}_\alpha|\text{unlearn}). \qquad (2)$$

This score isolates the video's specific contribution to knowledge recovery. A higher $\widetilde{S}$ indicates stronger knowledge transfer from the generated video.

**Token Cost & Time (Efficiency).** Beyond quality, we also assess the practical efficiency of educational video creation. To assess scalability, we report *average code generation time* and *token usage per video*. These metrics are crucial for real-time educational applications where latency and cost are practical constraints.

## 4. Method: Code2Video

**Overview.** Given a topic query $\mathcal{Q}$, Code2Video outputs a rendered video $\mathcal{V}$. As shown in Fig. 4, the pipeline comprises three agents: **(i) Planner** structures topics into storyboards with reference assets, **(ii) Coder** translates each section into executable Manim code using parallel synthesis and an effective debugging, and **(iii) Critic** refines rendered videos through a novel visual prompt and VideoLLM feedback to ensure spatial coherence and educational clarity.

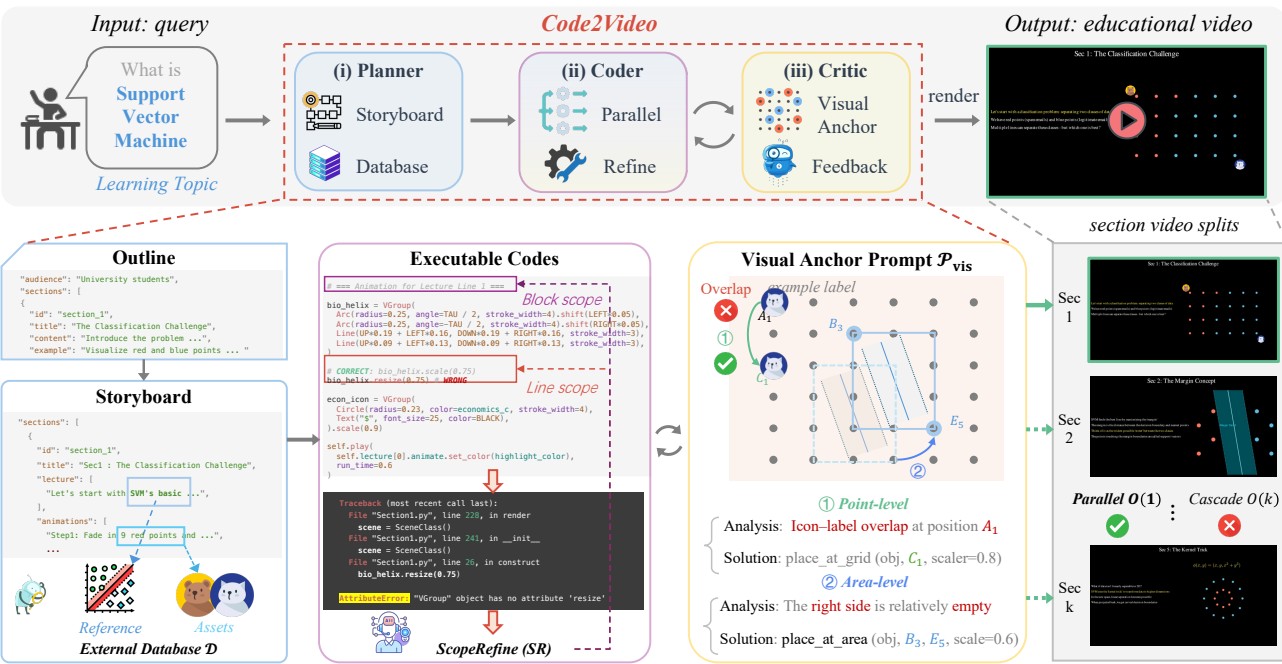

*Figure 4.* **Illustration of Code2Video.** Given a topic inquiry, Code2Video aims to render an educational video via Manim code writing: **(i) The Planner** constructs a hierarchical storyboard and retrieves semantic assets; **(ii) Coder** synthesizes section-wise code in parallel and applies **ScopeRefine** to locate and fix local bugs efficiently; **(iii) The Critic** enforces spatial coherence using *Visual Anchor Prompting*, which map continuous layouts to discrete grid coordinates for precise refinement.

## 4.1. Planner: query to storyboard

**(i) Outline Generation.** Given $\mathcal{Q}$, the Planner produces an outline $\mathcal{O} = o_1, \ldots, o_n$, where each $o_i$ contains section title, content summary, and illustrative examples. It tailors the structure to the intended audience (*e.g.,*trigonometric functions for middle school, Fourier's law for undergraduates), ensuring level-appropriate structure. Formally, $\mathcal{O} \leftarrow \mathcal{P}_{\text{outline}}(\mathcal{Q})$, where $\mathcal{P}_{\text{outline}}$ guides the LLM to produce coherent section metadata, establishing the temporal skeleton for the video.

**(ii) Storyboard Construction.** The second stage converts the outline $O$ into a detailed storyboard $s$. Each section in $s$ includes title, lecture lines, and corresponding animations, generated via $s_i \leftarrow \mathcal{P}_{\text{storyboard}}(o_i)$. The prompt $\mathcal{P}_{\text{storyboard}}$ directs the LLM to expand the outline into step-by-step visual scripts. The storyboard specifies the temporal sequence of lecture lines and paired animations, bridging high-level planning with concrete visual content.

**External Database.** To enhance factual accuracy and visual fidelity, the Planner retrieves assets from an external database $\mathcal{D}$. This includes *(a) reference images* aligned with the topic to **anchor complex concepts and reduce hallucination**, and *(b) reusable visual assets* (*e.g.,*icons, logos) that are difficult to generate from scratch. A prompt $\mathcal{P}_{\text{asset}}$ analyzes the storyboard to automatically identify required assets $\mathcal{A}$, via $a_i \leftarrow \mathcal{P}_{\text{asset}}(s_i)$. These are stored in a persistent cache $\mathcal{D}_{\text{asset}}$, enabling reuse across sections and ensuring visual consistency. Please refer to § A.1.6 for more details and examples about $\mathcal{D}$.

## 4.2. Coder: Storyboards to Executable Code

The Coder $\mathcal{G}$ translates each section of the storyboard $s$ and the cached assets $A$ into executable Manim code $C = \{c_1, \ldots, c_n\}$, where each $c_i$ corresponds to a storyboard $s_i$.

**(i) Parallel Code Synthesis.** The primary bottleneck is generation time: serial processing and error-prone code requiring LLM rewrites can extend generation to over 2 hours for a simple video. We address this by parallelizing the pipeline, handling each section independently via $c_i \leftarrow \mathcal{P}_{\text{coder}}(s_i, \mathcal{A})$. Here, $\mathcal{P}_{\text{coder}}$ guides the LLM to translate storyboard descriptions into executable Manim code. Shared assets $\mathcal{A}$ maintain temporal consistency across sections while preserving parallelization efficiency.

**(ii) ScopeRefine: Effective Debugging.** Even strong LLMs seldom generate fully executable code in one attempt. Basic repair strategies that concatenate entire code sections with full error logs incur substantial time and token costs. We propose **ScopeRefine (SR)**, a hierarchical repair strategy that escalates context only when needed, as illustrated in Fig.4 bottom center. *(a) Line scope.* Extract the error line with minimal context $\mathcal{S}_1$ (e.g., line±1) and attempt up to $K_1$ fixes. *(b) Block scope.* If the error persists, expand to

the corresponding lecture-line block $\mathcal{S}_2 = \mathcal{B}_{i,j}$ and attempt up to $K_2$ fixes. *(c) Global scope.* As a last resort, regenerate the full section $c_i$ from $s_i$. This progressive, "escalate-on-failure" design reduces latency and token cost while maintaining high success rates.

## 4.3. Critic: Anchor-guided Visual Refinement

Even after debugging ensures executability, the generated code may still yield unsatisfactory visual outcomes. LLMs and VLMs often fail to provide actionable feedback due to **limited spatial awareness** (Cheng et al., 2024; Zha et al., 2025). In practice, models can identify issues (*e.g.,*"the cat icon is misplaced") but struggle to provide actionable corrections. They often fail to indicate the direction or distance needed to adjust the element, which makes text-only refinement inadequate.

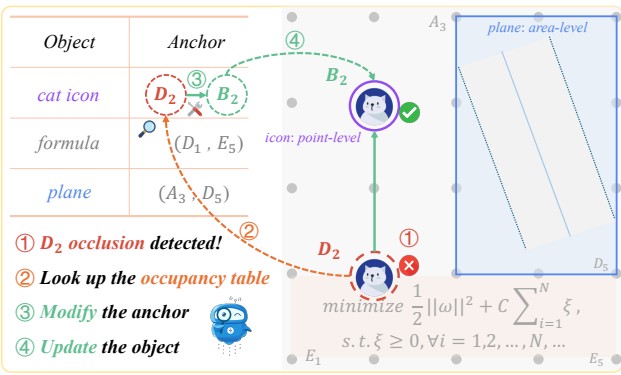

*Figure 5.* Illustration of *Visual Anchor Prompting* ($\mathcal{P}_{\mathrm{vis}}$).

**(i) Visual Anchor Prompting** ($\mathcal{P}_{\mathrm{vis}}$). We introduce $\mathcal{P}_{\mathrm{vis}}$ to make layout feedback directly executable. $\mathcal{P}_{\mathrm{vis}}$ discretizes the 2D canvas into a $6 \times 6$ grid of *anchor points*, where each cell is mapped to fixed Manim coordinates. The Critic expresses layout edits by selecting anchors, which can be translated into code without free-form geometry. Placement follows two granularities, as illustrated in Fig. 5: *(a) point-level*, for small elements (*e.g.,*symbols, short labels) occupy a single anchor; and *(b) region-level* for larger elements assigned to bounding boxes spanning multiple anchors. By discretizing the problem, we convert **continuous positioning** into a **discrete anchoring task**. This creates a visual "*go-to*" shortcut that substantially reduces the difficulty for LLMs to produce valid layouts.

**(ii) VideoLLM Feedback with Occupancy Tracking.** To detect violations and refine placement, the Critic inspects the rendered video $\mathcal{V}_i$ alongside its section code $c_i$. During parallel code generation, we maintain an *occupancy table* that records, for each visual element, its assigned anchor(s) (point/region), scale, and the code lines that instantiate it. This enables (a) fast tracing from a visual issue to the responsible code span, and (b) conflict-aware relocation by reveal-

ing available anchors. With this structure, the Critic focuses on common layout failures: overlap within a cell, narration occluded by graphics, and severe imbalance due to unused regions. These findings are fed into a refinement prompt $\mathcal{P}_{\mathrm{refine}}$ to produce updated code: $\tilde{c}_i \leftarrow \mathcal{P}_{\mathrm{refine}}(c_i, \mathcal{V}_i)$, and the final video is rendered as $\widetilde{\mathcal{V}} = \mathrm{Render}(\{\tilde{c}_i\}_{i=1}^n)$. By combining anchor-based edits, occupancy-aware constraints, and multimodal feedback, the Critic yields clearer and more stable layouts than text-only refinement.

## 5. Experiment

### 5.1. Implementation Details

**Baselines.** We compare four approaches: **(i) Human-crafted**: expert-designed Manim videos as an upper bound. **(ii) Pixel-based diffusion**: text-to-video models including *OpenSora-v2* (Peng et al., 2025), *Wan2.2-T2V-A14B* (Wan et al., 2025), and *Veo3* (Google DeepMind, 2025). **(iii) CodeLLM Creation**: direct Manim code generation from learning topics using LLMs. **(iv) Agentic Creation (ours)**: our Planner–Coder–Critic pipeline.

**Model Configuration.** We instantiate these methods with *Claude Opus 4.1* (Anthropic, 2025), *GPT-4o, GPT-o4 mini, GPT-4.1, GPT-5* (OpenAI, 2025), and *Gemini-2.5 Pro* (Imran & Almusharraf, 2024). Unless otherwise specified, *Gemini-2.5 Pro* serves as the Critic for refinement. We use *Gemini-2.5 Pro* as the VLM-as-a-Judge for aesthetics and compute knowledge transfer with TeachQuiz.

**Resources.** Reference images are retrieved via Google Images, and visual assets are sourced from Iconfinder and Freepik. All prompts are provided in §A.2.

### 5.2. Main Results

**Quantitative Results.** Table 1 compares Code2Video with human-crafted videos, pixel-based models, and code LLM baselines across Efficiency, Aesthetics (AES), and knowledge transfer (TeachQuiz). Our analysis reveals four key findings: **(i) Pixel-based models underperform.** Pixel-based models obtain the lowest AES and TeachQuiz scores, with near-zero performance on *Element Layout* and *Logic Flow*. This suggests that, for educational videos, the main challenge is not short-term visual plausibility, but maintaining symbolic precision, coherent exposition, and cross-frame consistency. **(ii) Code-centric generation delivers clear improvements**. Even direct Manim code generation substantially outperforms pixel-based baselines, improving AES from 3.7–9.0 to 32.7–37.8 and TeachQuiz from 0.0–2.5 to 36.5–40.0. This confirms that executable code is a much stronger intermediate representation for controllable and semantically faithful educational video creation. **(iii) Our agentic framework enables consistent gains.** Across all backbone LLMs, Code2Video consistently improves

*Table 1.* **Results across Efficiency, Aesthetics, and TeachQuiz.** Efficiency: Duration (**video duration**), Time (**avg generation minutes**) and Token (avg **token consumption**). Aesthetics: **E**lement **L**ayout, **AT**tractiveness, **L**ogic **F**low, **V**isual **C**onsistency, **A**ccuracy & **D**epth.

| Method | Efficiency (↓) | | | Aesthetics (↑) | | | | | | TeachQuiz (↑) |
|---|---|---|---|---|---|---|---|---|---|---|
| | Duration | Time (min) | Token (K) | EL | AT | LF | VC | AD | **Avg** | |
| Human-made 3B1B | 16.9 min | – | – | 98.3 | 100 | 100 | 100 | 100 | 99.7 | 97.1 |
| | | | *Pixel-based Diffusion* | | | | | | | |
| OpenSora-v2 | | 27.6 | – | 0.0 | 5.0 | 0.0 | 0.0 | 13.3 | 3.7 | 0.0 |
| Wan2.2-T2V-A14B | ∼ 8 s | 17.4 | – | 0.0 | 10.0 | 0.0 | 0.0 | 20.0 | 6.0 | 0.0 |
| Veo3 | | 2.3 | – | 0.0 | 15.0 | 0.0 | 5.0 | 25.0 | 9.0 | 2.5 |
| | | | *Code LLM* | | | | | | | |
| GPT-5 | | 1.8 | 1.1 | 27.0 | 28.0 | 28.0 | 54.5 | 26.0 | 32.7 | 36.5 |
| GPT-4.1 | ∼ 30 s | 2.1 | 1.2 | 30.5 | 34.5 | 39.0 | 42.0 | 24.8 | 34.2 | 37.0 |
| Claude Opus 4.1 | | 2.8 | 2.3 | 47.5 | 40.0 | 26.5 | 56.6 | 18.4 | 37.8 | 40.0 |
| | | | *Code2Video Agent (Ours)* | | | | | | | |
| Code2Video Gemini-2.5 Pro | | 15.5 | 41.8 | 70.3 | 60.3 | 44.3 | 37.6 | 74.7 | 57.4 | 72.0 |
| Code2Video GPT-4o | | 14.1 | 32.7 | 70.3 | 58.3 | 54.6 | 48.5 | 68.3 | 60.0 | 44.0 |
| Code2Video GPT-o4 mini | ∼ 2 min | 16.8 | 49.2 | 77.0 | 52.8 | 73.0 | 57.2 | 79.0 | 67.8 | 48.5 |
| Code2Video GPT-5 | | 8.8 | 19.3 | 75.5 | 60.5 | 81.8 | 63.6 | 79.7 | 72.2 +39.5 | 80.0 +43.5 |
| Code2Video GPT-4.1 | | 15.4 | 30.8 | 82.8 | 65.6 | 95.0 | 68.0 | 83.7 | 79.0 +44.8 | 82.0 +45.0 |
| Code2Video Claude Opus 4.1 | | 13.8 | 43.1 | 90.6 | 79.7 | 93.3 | 84.2 | 91.9 | **87.9** +50.1 | **86.0** +46.0 |

over direct Code LLM generation. With Claude Opus 4.1, AES rises from 37.8 to 87.9 and TeachQuiz from 40.0 to 86.0, i.e., gains of **+50.1** and **+46.0** points. These gains are especially pronounced on *Element Layout*, *Logic Flow*, and *Accuracy & Depth*, indicating that the framework improves not only visual quality but also instructional effectiveness. **(iv) Human-made videos remain the gold standard.** Human-made 3B1B videos still achieve the strongest results, reflecting advantages in storytelling, pacing, and explanatory nuance. Nevertheless, the best Code2Video configuration narrows the gap substantially, suggesting that the next frontier is closing the final distance to ***professional-quality, long-form educational video generation***.

**Generation Efficiency vs. Pixel-based Models.** A direct wall-clock comparison between pixel-based and code-centric methods is misleading because the two paradigms produce outputs of fundamentally different durations. Pixel-based models generate clips of ∼8 seconds, whereas *Code2Video* produces minute-level educational videos. To enable a fair comparison, we normalize generation cost by output video duration ($min_{cost}$ / $min_{video}$); lower values indicate higher efficiency. As shown in Table 2, despite producing **14–15×** **longer** videos, our code-centric approach achieves **2.5–43×** **better** normalized efficiency than all pixel-based alternatives.

**Qualitative Analyses.** Fig. 6 illustrates that our code-driven pipeline produces videos with clear text and formulas, stable layouts without occlusions, and stepwise alignment with lecture lines. In contrast, the pixel-based model (Veo3) often generates blurry or corrupted text, inconsistent styles,

*Table 2.* Efficiency comparison. *Eff.* = Cost (min) / Duration (min); lower is better (↓).

| Method | Type | Dur. | Eff. (↓) |
|---|---|---|---|
| OpenSora-v2 | Pixel-based | 8 s | 212.3 |
| Wan2.2-T2V-A14B | Pixel-based | 8 s | 133.8 |
| Veo3 | Pixel-based | 8 s | 17.3 |
| Code2Video GPT-5 | Code-centric | 1.8 m | **4.9** |
| Code2Video Claude Opus 4.1 | Code-centric | 2.0 m | **6.9** |

and drifting visuals, weakening semantic grounding. Overall, code-driven synthesis ensures better spatial stability and clearer knowledge presentation. Additional cases are provided in § A.1.7.

*Table 3.* Effect of different components on quality: TeachQuiz / Aesthetics avg. score.

| Method | Aesthetics | TeachQuiz |
|---|---|---|
| Code2Video Chat-4.1 (◇) | **79.0** | **82.0** |
| ◇ w/o Planner | 38.1 −40.9 | 40.5 −41.5 |
| ◇ w/o External Database | 68.1 −10.9 | 52.0 −30.0 |
| ◇ w/o Visual Anchor | 69.2 −9.8 | 55.2 −26.8 |
| ◇ w/o Critic | 72.5 −6.5 | 60.7 −21.3 |

### 5.3. Ablation Studies

**Effects by Individual Components.** Table 3 highlights several key patterns. First, TeachQuiz is more sensitive than Aesthetics, revealing *knowledge-transfer gaps even when videos remain visually acceptable*. Second, the Planner is essential: its removal causes both metrics to drop substantially (≈ 41 points), underscoring that high-level lecture

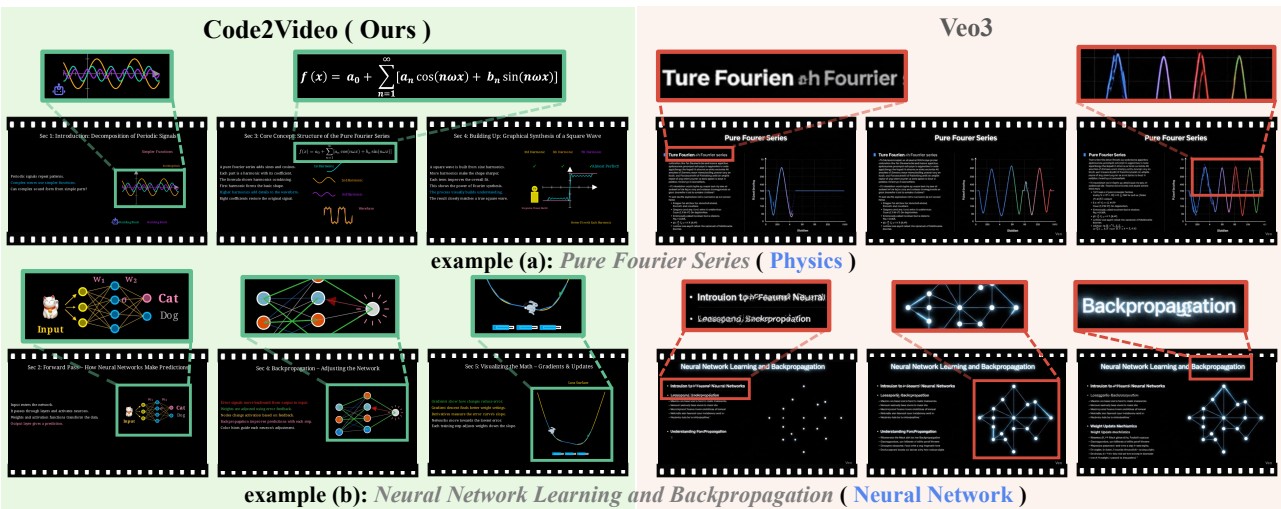

Figure 6. Qualitative comparison between *Code2Video* and *Veo3*. Our approach generates videos with coherent logic flow, consistent semantics, and interpretable layouts.

planning and temporal sequencing form the foundation of effective educational videos. Third, other modules provide complementary gains: the External Database improves conceptual grounding, Visual Anchors stabilize layouts, and the Critic ensures refinement—all contributing to the pipeline's robustness. These results **highlight that structured visual guidance and iterative refinement are crucial** for producing visually clear videos that effectively convey knowledge.

*Table 4.* **Ablation of efficiency components.** Average runtime (minutes) and token usage (K).

| Method | Time (min) | Token (K) |
|---|---|---|
| Code2Video **Chat-4.1** ($\diamond$) | **15.4** | **30.8** |
| $\diamond$ w/o parallel | 86.6 $_{5.6\times}$ | 30.8 |
| $\diamond$ w/o SR $\rightarrow$ w. Retry | 42.9 $_{2.8\times}$ | 49.8 $_{1.6\times}$ |
| $\diamond$ w/o SR $\rightarrow$ w. Debug | 39.2 $_{2.5\times}$ | 42.1 $_{1.4\times}$ |
| $\diamond$ w/o parallel & SR | 149.8 $_{9.7\times}$ | 52.6 $_{1.7\times}$ |

*Table 5.* **Re-evaluation with alternative VLM judges.** AES and TQ report average scores; Kendall $\tau$ measures rank correlation with the default judge.

| Judge | AES / TQ ($\uparrow$) | | $\tau$ |
|---|---|---|---|
| | CodeLLM | Code2Video | |
| Gemini-2.5 Pro **(default)** | 37.8 / 40.0 | 87.9 / 86.0 | — |
| GPT-5 | 36.2 / 38.0 | 85.1 / 83.5 | 0.93 |
| QwenVL-3 | 35.0 / 36.5 | 83.7 / 81.2 | **0.89** |

**Impact of Efficiency Components.** Table 4 evaluates efficiency-oriented modules. Parallelization is critical for temporal scalability; ablating it forces serial execution, causing a severe latency spike ($15.4 \rightarrow 86.6$ min, a $5.6\times$ **slowdown**). **ScopeRefine (SR)** demonstrates the advantage of

fine-grained repair over coarse-grained baselines. Replacing SR with *Naive Retry* (regenerating the full section on error) or *Global Debug* (regenerating with full error logs) incurs significant computational redundancy, increasing token costs by $1.6\times$ and $1.4\times$, respectively. Removing both mechanisms results in prohibitive overheads ($\sim$2.5 hours per video), confirming that parallel synthesis and localized repair are prerequisites for scalable, code-centric creation.

**Ablation on Unlearning Methods.** We compare a fine-tuning unlearning method (Yuan et al., 2024a) against training-free in-context unlearning (Thaker et al., 2024) and ours. For fine-tuning, we follow (Yuan et al., 2024a) by selecting 20 topics for the forget set and 20 for the neighbor set, and fine-tune *llava-pretrain-llama-2-7b-chat* (Liu et al., 2023). We then generate Code2Video videos from the forget set and compute TeachQuiz on these videos under each unlearning method. Table 7 reveals two findings. **(i) Efficiency.** On LLaVA, the training-free approach **performs comparably** to fine-tuning while avoiding the high cost of training. **(ii) Clearer TeachQuiz.** Training-free methods enable the use of stronger closed-source models. While fine-tuning is restricted to weaker open-source VLMs (yielding smaller learning gains), in-context unlearning leverages powerful models to demonstrate clearer post-video improvement.

**Evaluator Independence.** Our pipeline uses Gemini-2.5 Pro as both the Critic and the VLM-as-Judge, which raises the question of whether aesthetic scores reflect genuine quality or evaluator bias. Crucially, the two roles operate on fundamentally different representations: the Critic reads and edits *source code* via anchor-based spatial feedback, while the Judge scores *rendered video frames* across five perceptual axes post hoc — with no shared intermediate

*Table 6.* **Human study** on Aesthetics, TeachQuiz, Completion Willingness (CW), and Average Ranking (AR). Results align with VLM-based trends but show sharper score contrast, lower tolerance for layout errors, and reduced engagement in longer-duration videos.

| Method | Duration | Aesthetics (↑) | | | | | | TeachQuiz (↑) | CW (↑) | AR (↓) |
| | | EL | AT | LF | VC | AD | Avg | | | |
|---|---|---|---|---|---|---|---|---|---|---|
| Human-made 3B1B | 16.9 min | 98.9 | 97.2 | 91.3 | 98.0 | 97.0 | 96.5 | 78.8 | 36.2 | 1.2 |
| Pixel-based Veo3 | 8.0 s | 12.6 | 4.4 | 1.1 | 24.4 | 1.1 | 8.5 | 8.0 | 46.8 | 5.0 |
| Code LLM Claude Opus 4.1 | 0.5 min | 16.1 | 41.1 | 55.6 | 71.1 | 72.2 | 51.2 | 56.6 | 15.0 | 3.9 |
| Code2Video Gemini-2.5 Pro | 1.6 min | 26.7 | 68.3 | 78.1 | 90.2 | 81.0 | 68.9 | 65.3 | 47.4 | 3.1 |
| Code2Video Claude Opus 4.1 | 2.0 min | 60.2 | 89.3 | 84.6 | 92.0 | 83.1 | **81.8** | **80.3** | **64.0** | 1.8 |

*Table 7.* **Comparison of unlearning methods for TeachQuiz.** For each evaluation VLM, we apply different unlearning methods and compute TeachQuiz on videos rendered from Manim code generated by Claude Opus 4.1, GPT-5, and GPT-4o.

| Evaluation VLM | Unlearning Method | Video Source Model | | |
| | | Opus 4.1 | GPT-5 | GPT-4o |
|---|---|---|---|---|
| LLaVA-Next | Fine-tuning | 30 | 24 | 15 |
| | In-Context | 32 | 25 | 16 |
| | **Ours** | 31 | 20 | 13 |
| GPT-5 | Fine-tuning | *Not Applicable* | | |
| | In-Context | 70 | 61 | 33 |
| | **Ours** | 72 | 61 | 36 |
| Gemini-2.5 Pro | Fine-tuning | *Not Applicable* | | |
| | In-Context | 81 | 79 | 38 |
| | **Ours** | 84 | 77 | 42 |

state. TeachQuiz is computed independently via quiz accuracy and is unaffected by either component. To empirically verify this independence, we re-score all methods using two held-out judges unseen during development. As shown in Table 5, rank ordering remains stable across all three judges (Kendall $\tau \geq 0.89$), confirming that the observed performance gains reflect properties of the generated videos rather than idiosyncrasies of any single evaluator.

**Human Study Evaluation.** We conduct a five-group user study with 6 middle school and 2 undergraduate volunteers per group. Each participant watches one video type and answers 5 quiz questions across 20 learning topics. We measure Completion Willingness (**CW**, proportion finishing the video before answering, max score 100) and Average Ranking (**AR**, mean preference across video types, 1 is best). Table 6 reveals four patterns. **(i) Clearer separation.** Human ratings follow the same trends as VLM-based scores but with sharper contrast: high-quality videos score above 90 while low-quality videos fall below 10. **(ii) Sensitivity to layout errors.** Participants assign notably lower Element Layout scores (EL) to Code2Video outputs, reflecting a higher human sensitivity to even brief occlusions — issues that VideoLLMs frequently overlook. **(iii) Attention span limits.** Effective quiz performance requires participants to sustain attention throughout the full video. This demands

not only *strong logical coherence* and *engaging presentation*, but also a *reasonable duration* that supports continuous knowledge absorption without inducing fatigue. **(iv) Strong consistency.** Human scores for Aesthetics and TeachQuiz are highly correlated, indicating that visual appeal and learning efficacy are mutually reinforcing. Overall, the human study confirms that structural clarity and visual appeal are both crucial for learning efficacy, complementing our automated metrics. Future work should develop agent designs that explicitly account for **human attention and patience**, producing videos that preserve **fine-grained detail** while **minimizing perceptual fatigue**.

## 6. Conclusion

In this work, we propose a code-centric paradigm for educational video creation, where executable programs serve as the medium for temporal sequencing and spatial organization. On top of this paradigm, our tri-agent framework *Code2Video* enables controllable and interpretable generation with multimodal feedback. To support systematic evaluation, we introduce *MMMC*, a benchmark that measures efficiency, aesthetics, and knowledge conveyance. Together, our paradigm, framework, and benchmark provide a basis for future research on using code to generate high-quality, structured, and interpretable educational content.

**Future Work.** We identify three directions for extending this work. **(i) Broader rendering backends.** The Planner, Critic, and TeachQuiz components are rendering-engine agnostic by design. Adapting the Coder to alternative backends (e.g., Blender, Unity) would extend the paradigm to 3D environments and organic structures without altering the core architecture. **(ii) Hybrid code–pixel generation.** Code-centric and pixel-based generation are complementary: the former offers structured symbolic control, while the latter provides photorealistic visual richness. A natural extension is to treat pixel generation as a callable module within the Coder, while retaining code as the orchestration layer for temporal structure and spatial coherence. **(iii) Efficient agent design.** Future gains may come from lighter specialized agents, speculative code generation, and stronger debugging oracles, moving toward low-latency, on-demand educational video creation.

## Acknowledgements

This research is supported by the National Research Foundation, Singapore under its AI Singapore Programme (AISG Award No: AISG3-RP-2022-030).

## Impact Statement

This work introduces a code-centric framework for generating educational videos and an evaluation protocol for measuring knowledge conveyance. A positive impact is improved access to high-quality instructional content: code-based generation is controllable, interpretable, and easy to edit, which can reduce production cost and support rapid customization for different audiences, languages, and learning needs. The proposed benchmark and metrics may also enable more rigorous and reproducible research on educational video generation.

This technology also carries risks. The approach can be misused to scale the production of persuasive but incorrect content. In addition, evaluation with large VLMs may inherit biases and may not fully reflect human learning outcomes.

We mitigate these risks by emphasizing transparency and auditability: outputs are executable programs that can be inspected and corrected, and our evaluation is reported alongside human studies rather than treated as a standalone guarantee. We encourage future work to incorporate stronger fact-checking, domain expert review for high-stakes topics, and broader user studies across diverse learner groups.

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

# A. Appendix

## A.1. Additional Implementation Details and Experiments

### A.1.1. UNLEARNING DETAILS AND TEACHQUIZ

To probe whether generated educational videos genuinely transfer knowledge, we integrate a selective unlearning–relearning protocol into the TeachQuiz evaluation.

**Model choice.** We adopt *Gemini-2.5 Pro* (Imran & Almusharraf, 2024), one of the current state-of-the-art models in video understanding. Its closed-source nature precludes parameter-level interventions for unlearning; thus, we rely on a prompt-based strategy, a standard approach for steering proprietary models.

**Unlearning stage.** We design a parameter-free pipeline $\mathcal{P}_{\text{unlearn}}$ tailored for closed-source models. Given a target concept $\mathcal{K}$, we define a shadow knowledge set $\mathcal{B}(\mathcal{K})$ consisting of canonical definitions, formulas, aliases, and exemplars associated with $\mathcal{K}$. During inference, $\mathcal{P}_{\text{unlearn}}$ enforces: (i) *contextual masking*, where $\mathcal{B}(\mathcal{K})$ is silently identified and treated as inaccessible; (ii) *uncertainty injection*, where the model must output "*INSUFFICIENT EVIDENCE*" whenever the reasoning chain depends on elements of $\mathcal{B}(\mathcal{K})$; (iii) *progressive forgetting validation*, where queries of increasing difficulty $\{q_i\}_{i=1}^{N}$ are used to test suppression not only at recall-level but also across multi-step reasoning. Formally, the model's answer distribution is constrained to

$$f(q_i \mid \mathcal{P}_{\text{unlearn}}) \in \{y_i, \text{NULL}\}, \tag{3}$$

where NULL indicates blocked inference. This layered design obstructs both direct recall and indirect reconstruction, ensuring that performance degradation reflects genuine unlearning rather than prompt compliance artifacts.

**Relearning stage.** We then expose the model to an educational video $\mathcal{V}$ and apply a relearning prompt $\mathcal{P}_{\text{learn}}$, which restricts evidence scope to $\mathcal{V}$ while maintaining the block on $\mathcal{B}(\mathcal{K})$. The answering constraint becomes

$$f(q_i \mid \mathcal{P}_{\text{learn}}, \mathcal{V}) \in \{y_i, \text{NULL}\}, \tag{4}$$

with justification required to reference only cues present in $\mathcal{V}$. This ensures that any gain after relearning is attributable solely to video-grounded evidence rather than residual prior knowledge.

**Evaluation setup.** For each learning topic, we construct 10 multiple-choice questions with four options (A–D), each containing exactly one correct answer. To better capture the expressive power of educational videos, these quizzes emphasize visually grounded reasoning. For instance, rather than simply asking *"What is the definition of a complex number?"*, a question may ask *"When a point moves on the complex plane, what visual transformation corresponds to multiplication by $i$?"*. Such queries demand alignment between knowledge and its visual instantiation.

**Metric.** Given a concept $\mathcal{K}$, we construct $N$ multiple-choice questions $\{q_i\}_{i=1}^{N}$ with ground-truth answers $\{y_i\}_{i=1}^{N}$. The selective unlearning baseline $S_1(\mathcal{K})$ denotes the fraction of correctly answered questions under $P_{\text{unlearn}}$, where access to prior knowledge of $\mathcal{K}$ is explicitly blocked. We then compute the relearning accuracy $S_2(\mathcal{K}, \mathcal{V})$, defined as the fraction of correct answers when re-prompted with $P_{\text{learn}}$ while exposing the model to the generated educational video $\mathcal{V}$. Formally,

The *TeachQuiz* score is then defined as:

$$\text{TQ}(\mathcal{K}, \mathcal{V}) = S_2(\mathcal{K}, \mathcal{V}) - S_1(\mathcal{K}),$$

which captures the relative gain in accuracy attributable solely to $\mathcal{V}$. Intuitively, $S_1$ reflects how well the model resists using forbidden prior knowledge, while $S_2$ reflects how much can be recovered from the video. A higher TQ thus indicates stronger video-induced knowledge acquisition.

**Ablation on evidence sources.** To ensure that the observed gains are indeed attributable to the generated videos, we conduct an ablation study, shown in Table 8.

First, when providing only **Text-only** lecture lines (akin to PDF-style slides without animation), performance improves moderately compared to the unlearn baseline but falls short of full video-based relearning, highlighting that textual scaffolding alone is insufficient.

Second, with **Animation-only** inputs (animations without accompanying lecture text), accuracy also rises above unlearn but remains lower than the full condition, suggesting that temporal visual cues contribute substantially but require textual grounding for maximum effect.

*Table 8.* Ablation on unlearning. Accuracy reports correct concept judgments; $\Delta = \text{TQ}$ denotes the improvement in TeachQuiz confidence from the Unlearn setting to the Relearn setting. Text-only/Animation/Random evaluate TeachQuiz (TQ) under partial or mismatched supervision.

| Method | Accuracy | | | TeachQuiz (TQ) | | |
|---|---|---|---|---|---|---|
| | Unlearn | Relearn | $\Delta = \text{TQ}$ | Text-only | Animation | Random |
| Code2Video GPT-5 | 5.0 | 85.0 | 80.0 | 27.2 | 72.1 | 2.0 |
| Code2Video GPT-4.1 | 5.0 | 87.0 | 82.0 | 22.1 | 75.0 | 5.0 |
| Code2Video Claude Opus 4.1 | 5.0 | 91.0 | 86.0 | 24.0 | 76.6 | 4.0 |

Finally, in the **Random-video** setting, where the VLM is paired with an unrelated topic video, performance collapses to the unlearn level (or lower), confirming that improvements do not stem from superficial video exposure but rather from semantically aligned educational content.

Overall, these results provide evidence that the generated videos drive knowledge reacquisition: text and animation are complementary, and their synergy yields the strongest TeachQuiz gains.

### A.1.2. HUMAN STUDY: MIDDLE SCHOOL VS. UNDERGRADUATE COMPARISON

Table 9 compares middle school and undergraduate participants on Aesthetics, TeachQuiz, and Completion Willingness (CW). As TeachQuiz measures knowledge acquisition, middle school students—closer to a true "unlearned" state—benefit more from effective videos, showing substantial TeachQuiz gains (e.g., Code2Video boosts middle school TeachQuiz to 88.1 versus 55.0 for undergraduates). Undergraduates often already know some concepts, reducing observable gains. Across both groups, Code2Video achieves high Aesthetics and CW, outperforming pixel-based models by large margins. Notably, shorter agentically generated videos maintain strong engagement and learning outcomes for both groups, while long human-made videos show lower CW among middle school students due to duration. Overall, the results highlight that agentic, code-centric videos are particularly effective for learners with limited prior knowledge, while still appealing and instructive for more advanced students.

*Table 9.* Comparison of middle school and undergraduate participants on Aesthetics, TeachQuiz, and Completion Willingness (CW).

| Method | Duration | Middle School | | | Undergraduate | | |
|---|---|---|---|---|---|---|---|
| | | Aesthetics | TeachQuiz | CW | Aesthetics | TeachQuiz | CW |
| Human-made 3B1B | 16.9 min | 96.3 | **86.3** | 34.9 | 97.5 | 56.0 | 40.2 |
| Pixel-based Veo3 | 8.0 s | 10.7 | **6.0** | 55.6 | 2.0 | 14.0 | 20.5 |
| Code2Video Claude Opus 4.1 | 5.0 min | 81.7 | **88.1** | 76.0 | 82.2 | 55.0 | 58.2 |

### A.1.3. ABLATION ON VISUAL ANCHOR POINT GRANULARITY

*Table 10.* Ablation on **anchor point granularity** in the Visual Anchor Point ($\mathcal{P}_{\text{vis}}$) design. Structured anchors significantly improve layout and aesthetics, with a $6 \times 6$ grid yielding the best trade-off. Finer grids (e.g., $8 \times 8$) cause clutter, while unconstrained (Self-directed) placement underperforms due to inconsistent spacing. **EL** stands for Element Layout, and **AT** stands for Attractivenss.

| # Anchor Points | AES | | | AES Avg. |
|---|---|---|---|---|
| | EL | AT | ( EL + AT ) / 2 | |
| w/o $\mathcal{P}_{\text{vis}}$ | 45.2 | 54.7 | 50.0 | 69.2 |
| Center Point | 49.0 | 56.4 | 52.7 | 69.7 |
| $4 \times 4$ | 76.1 | 63.0 | 69.6 | 76.9 |
| $6 \times 6$ | 82.8 | 65.6 | **74.2** | **79.0** |
| $8 \times 8$ | 77.2 | 60.6 | 68.9 | 76.0 |
| Self-directed | 48.8 | 57.3 | 53.1 | 70.3 |

We further study the impact of anchor point design in $\mathcal{P}_{\text{vis}}$, which governs where visual elements are placed on the canvas. Table 10 reports results under the AES framework, focusing on Element Layout (EL) and Attractiveness (AT), the two most placement-sensitive dimensions.

*Table 11.* Comparison on TheoremExplainBench (Ku et al., 2025). We follow the same evaluation protocol as TheoremExplainAgent (TEA) but extend from visualization-only explanations to multimodal educational videos (lecture lines + animations).

| Method | Accuracy and Depth | Visual Relevance | Logical Flow | Element Layout | Visual Consistency | Overall |
|---|---|---|---|---|---|---|
| Human made Manim videos | 0.80 | 0.81 | 0.70 | 0.73 | 0.87 | 0.77 |
| TEA Gemini 2.0 Flash | 0.79 | 0.75 | 0.84 | 0.58 | 0.87 | 0.76 |
| TEA o3-mini | 0.76 | 0.76 | 0.89 | 0.61 | 0.88 | 0.77 |
| TEA GPT-4o | 0.79 | 0.79 | 0.89 | 0.59 | 0.87 | 0.78 |
| Code2Video Gemini 2.0 Flash | 0.81 | 0.80 | 0.92 | 0.88 | 0.70 | 0.82 |
| Code2Video o3-mini | 0.76 | 0.86 | 0.92 | 0.90 | 0.93 | 0.87 |
| Code2Video GPT-4o | 0.82 | 0.91 | 0.86 | 0.91 | 0.92 | **0.88** |

**Setup.** We compare six variants: (i) w/o $\mathcal{P}_{\text{vis}}$, i.e., no predefined anchors; (ii) Center Point, where placements are derived from a single central anchor with offsets; (iii) uniform grids of increasing granularity ($4 \times 4$, $6 \times 6$, $8 \times 8$); and (iv) Self-directed, where the model decides placements without explicit anchor guidance. All variants above are instantiated with ChatGPT-4.1.

**Findings.** Three observations emerge. (1) **Structured anchors substantially improve layout quality.** Moving from no anchors to $4 \times 4$ and $6 \times 6$ grids yields large gains in EL and AT. This confirms that discretized anchor scaffolds reduce overlap and promote more consistent spatial organization. (2) **Moderation is key.** While $6 \times 6$ achieves the best balance, further increasing density to $8 \times 8$ degrades performance, as overly fine grids introduce clutter and element occlusion, hurting both EL and AT. (3) **Unconstrained placement is suboptimal.** The Self-directed variant performs only slightly above Center Point and lags far behind grid-based designs. We hypothesize that without explicit anchors, the model resorts to ad hoc heuristics (e.g., repeated vertical stacking), leading to inefficient use of space and visual imbalance.

Overall, the results highlight that *anchor granularity acts as a structural prior*: moderate discretization (here, $6 \times 6$) provides sufficient flexibility while preventing crowding, thereby offering the best trade-off between precision and aesthetics.

### A.1.4. EVALUATION ON THEOREMEXPLAINBENCH

Beyond our primary benchmark, we further test Code2Video on *TheoremExplainBench* (Ku et al., 2025), originally proposed to evaluate LLMs' capacity for visualizing abstract mathematical concepts. Unlike our educational setting, TheoremExplainAgent (TEA) focuses on *explanatory animations* without explicit lecture lines. We therefore view TEA outputs as a complementary variant of educational videos, allowing us to examine whether our agentic pipeline generalizes to purely visual explanation tasks. Table 11 reports the results, and the comparison yields three key findings.

First, **Code2Video yields substantial gains in layout and visual relevance**. With GPT-4o, Element Layout improves from 0.59 (TEA) to 0.91, and Visual Relevance from 0.79 to 0.91, with consistent gains across backbones. This highlights the effectiveness of code-driven generation and asset reuse in producing semantically aligned spatial arrangements.

Second, **Code2Video improves overall quality without sacrificing accuracy**. Overall scores rise by 0.06–0.10 over TEA, while Accuracy & Depth remains comparable or better. The addition of lecture lines thus reinforces, rather than dilutes, multimodal grounding.

Third, **model-specific trade-offs remain**. For example, Gemini-2.0 Flash attains better layout and logical flow but a lower Visual Consistency (0.70 vs. 0.87). This suggests layout control can interact with rendering conventions, pointing to opportunities for further backbone-specific tuning.

These gains can be attributed to several design choices in Code2Video. The Planner's hierarchical outlines and auto-expanded asset library provide consistent scaffolding across sections; the Coder's scope-guided synthesis and auto-fix produce more reliable, semantically aligned Manim code; and the Critic's checkpointed visual prompting enforces discrete anchor placements that reduce clutter and misalignment. Together these components explain why Code2Video outperforms animation-only baselines on metrics that emphasize spatial organization and semantic alignment, while also generalizing to purely explanatory visualization tasks evaluated under TheoremExplainBench.

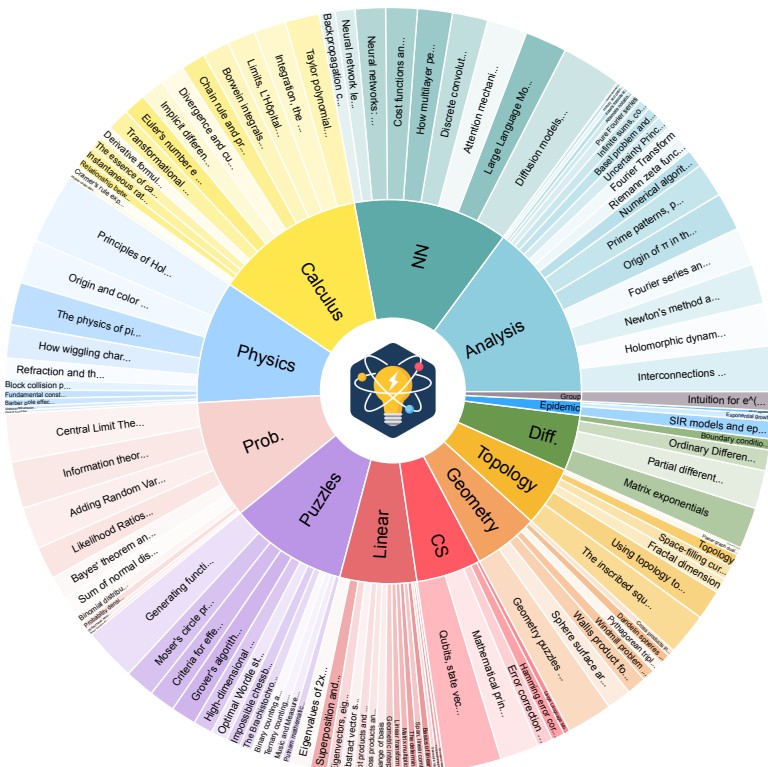

*Figure 7.* Distribution of 13 subject categories with exemplar learning topics. The width of the ring for each category represents the total duration of videos in that category.

### A.1.5. DETAILS OF MMMC

**Data Collection.** Our dataset targets A **M**assive **M**ulti-discipline **M**ultimodal **C**oding benchmark (**MMMC**) for code-driven educational video creation. Constructing a benchmark for code-driven educational video creation requires curating topics that are both pedagogically valuable and faithfully realizable in Manim code. Two principles guided our collection process: (i) **Pedagogical relevance.** Each educational topic should represent a concept with established teaching value, ensuring that generated videos are not synthetic artifacts but genuine instructional material. (ii) **Executable grounding.** Each educational topic must admit a high-quality reference video created by practitioners with substantial Manim expertise, guaranteeing that the underlying visualization is not only theoretically possible but also practically realizable. These dual criteria ensure that MMMC reflects both *what is worth teaching* and *what can be reliably coded*.

To satisfy these requirements, we turned to the **3Blue1Brown** (3B1B) repository [1], which uniquely balances pedagogical impact and Manim craftsmanship. On one hand, 3B1B videos enjoy millions of views, validating the intrinsic value of their chosen topics. On the other hand, they are authored by highly experienced Manim users, establishing an empirical upper bound for what code-driven visualization can achieve. Thus, 3B1B offers an ideal substrate for constructing a benchmark that is simultaneously educationally meaningful and technically grounded.

Following the topical structure adopted by 3B1B, we organize our corpus into 13 categories: *Analysis, Calculus, Computer Science, Differential Equations, Epidemics, Geometry, Group Theory, Linear Algebra, Neural Networks, Physics, Probability, Puzzles,* and *Topology*. From YouTube [2], we scraped the complete collection of 3B1B videos, then manually filtered out off-topic items such as Q&A sessions or non-instructional content, resulting in a curated set of 117 long-form videos.

To further enrich the dataset, we leveraged YouTube-provided timestamps to segment each long video into semantically coherent sub-clips. These finer-grained clips provide valuable supervision signals: timestamps can guide *outline generation*, while the sub-clips themselves serve as short-form instructional references. Finally, we distilled educational topics from

---

[1] https://www.3blue1brown.com/

[2] https://www.youtube.com/@3blue1brown/videos

both long videos and their sub-clips by prompting an LLM $\mathcal{P}_{\text{topic}}$ with titles, descriptions, and metadata, yielding a clean mapping from videos to pedagogically grounded knowledge units.

**Dataset Statistics.** Our curated dataset, MMMC, consists of a total of 456 educational videos, including 117 full-length videos and 339 timestamped segments. On average, a full-length video lasts 1014.41 seconds ($\sim$16.9 minutes), while a segmented clip spans 201.13 seconds ($\sim$3.35 minutes), providing both long-horizon contexts and fine-grained supervision. The extracted educational topics are concise yet precise, with an average length of 6.28 words per point. Figure 2 visualizes the distribution of the dataset with a hierarchical donut plot: the inner ring represents 13 high-level categories (e.g., *geometry*, *physics*, *topology*, *neural networks*), while the outer ring shows individual educational topics, where the arc width corresponds to the cumulative duration. This organization highlights both the topical diversity and the temporal richness of MMMC, making it a balanced and challenging benchmark for educational video creation.

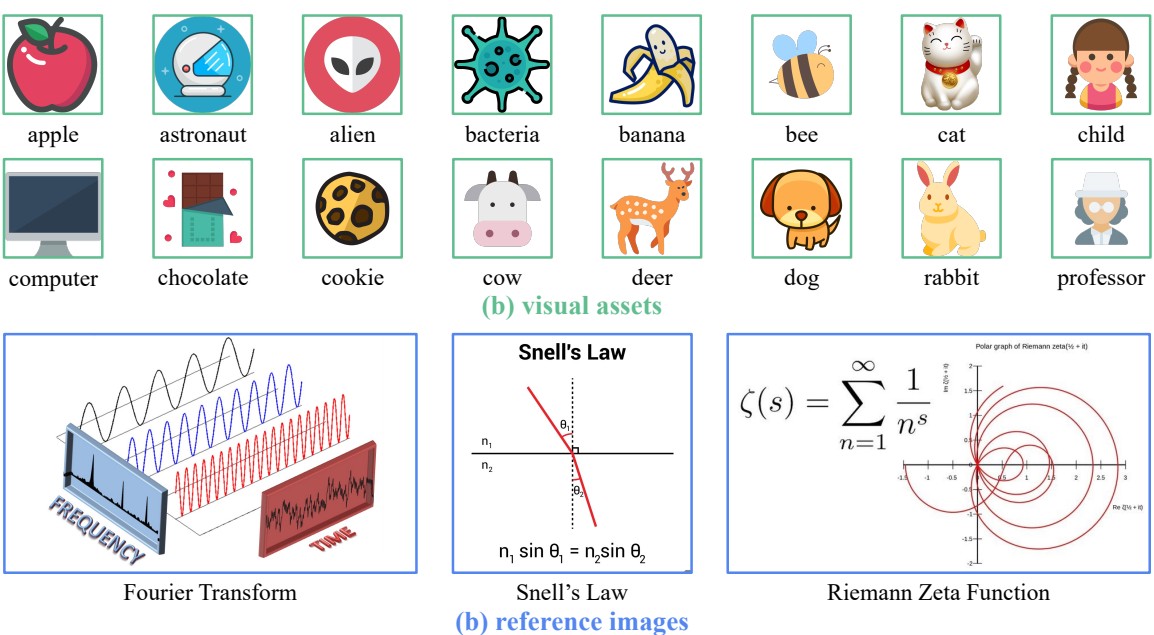

*Figure 8.* Sample reference images and visual assets from the external database, illustrating the types of visual materials used to enhance aesthetics, maintain consistency across sections, and support the depiction of complex concepts.

### A.1.6. EXTERNAL DATABASE

Figure 8 illustrates sample reference images and visual assets retrieved by our system. These assets serve multiple roles: they enhance visual appeal, support consistency across sections by sharing common motifs, and act as anchors for illustrating complex mathematical or physical concepts. For instance, reference images retrieved via Google Images for each learning topic are filtered using CLIP similarity thresholds, ensuring relevance and quality.

Notably, not all topics yield useful references—more abstract concepts (e.g., *Topology*) lack clear visual counterparts, limiting the benefit. Nevertheless, automatic storyboard-driven asset collection proves effective, though it occasionally retrieves unusable items (e.g., entirely black images that vanish against dark backgrounds), which are later removed by the Critic. Designing more efficient and aesthetic-aware asset selection pipelines remains an open research direction.

### A.1.7. QUALITATIVE ANALYSES

We provide qualitative case studies in Figure 9 and Figure 10. Figure 9 showcases generated videos across diverse learning topics, including *Euler's Formula*, *The Determinant*, *Pure Fourier Series*, *Space-filling Curves*, and *Neural Network Learning and Backpropagation*. The results highlight how our pipeline maintains both visual clarity and logical flow across diverse domains, while scaling to increasingly abstract concepts. Figure 10 further compares our approach with diffusion-based text-to-video models (*Veo3 (Google DeepMind, 2025)*, *Wan2.2-T2V-A14B (Wan et al., 2025)*) under the topics *The Determinant* and *Space-filling Curves*. Despite generating videos under 8s, diffusion models struggle with

text rendering, symbol precision, and fine-grained animations, producing outputs that are often visually inconsistent or pedagogically misleading. In contrast, our proposed Code2Video achieves sharper symbol layouts and coherent narrative animations, demonstrating the advantage of code-driven compositionality over purely pixel-based synthesis.

## A.2. Prompts of Code2Video

### A.2.1. PROMPT OF VLM-AS-JUDEGS FOR AESTHETICS

---

Prompt of VLM-as-judegs for aesthetics ($\mathcal{P}_{\mathrm{aesth}}$)

```
 1  You are an expert educational content evaluator specializing in instructional videos with synchronized
        presentations and animations. Please thoroughly analyze the provided educational video across five
        critical dimensions and provide detailed scoring.
 2
 3  EVALUATION FRAMEWORK:
 4
 5  1. Element Layout (20 points)
 6  Assess the spatial arrangement and organization of visual elements:
 7  - Clarity and readability of text/diagrams in the presentation (left side)
 8  - Optimal positioning and sizing of animated content (right side)
 9  - Balance between presentation and animation areas
10  - Appropriate use of whitespace and visual hierarchy
11  - Consistency in font sizes, colors, and element positioning
12  - Overall aesthetic appeal and professional appearance
13
14  2. Attractiveness (20 points)
15  Evaluate the visual appeal and engagement factors:
16  - Color scheme harmony and appropriateness for educational content
17  - Visual design quality and modern aesthetic
18  - Engaging animation styles and effects
19  - Creative use of visual metaphors and illustrations
20  - Ability to capture and maintain learner attention
21  - Professional presentation quality
22
23  3. Logic Flow (20 points)
24  Analyze the pedagogical structure and content progression:
25  - Clear introduction, development, and conclusion of concepts
26  - Logical sequence of information presentation
27  - Smooth transitions between topics and concepts
28  - Appropriate pacing for learning comprehension
29  - Coherent connection between presentation content and animations
30  - Progressive complexity building (scaffolding)
31
32  4. Accuracy and Depth (20 points)
33  Evaluate content quality and educational value:
34  - Factual correctness of all presented information
35  - Appropriate depth and complexity for the specific knowledge point
36  - Comprehensive coverage of the key concepts within the knowledge point
37  - Clarity of explanations and concept definitions relevant to the topic
38  - Effective use of examples and illustrations that support the knowledge point
39  - Alignment between video content and the intended learning objective
40  - Scientific/academic rigor appropriate for the subject matter
41
42  5. Visual Consistency (20 points)
43  Assess uniformity and coherence throughout:
44  - Consistent visual style across all elements
45  - Uniform color palette and design language
46  - Coherent animation styles and timing
47  - Consistent typography and formatting
48  - Smooth integration between static and animated elements
49  - Maintaining visual standards throughout the entire video
50
51  SCORING INSTRUCTIONS:
52  - Provide a score for each dimension (exact decimal allowed)
53  - Calculate overall score as sum
54  - Provide specific feedback for each dimension, considering the knowledge point context
55  - Evaluate whether the video effectively teaches the specified knowledge point
56  - Assess if the pedagogical approach is suitable for the subject matter
57  - Consider if animations and visual elements appropriately support the knowledge point
58
59  RESPONSE FORMAT:
60  MUST structure your response in the following JSON format:
61
62  {{
63  "element_layout": {{
```

---

```
64        "score": [0-20],
65        "feedback": "Detailed analysis of layout quality..."
66    }},
67    "attractiveness": {{
68        "score": [0-20],
69        "feedback": "Assessment of visual appeal..."
70    }},
71    "logic_flow": {{
72        "score": [0-20],
73        "feedback": "Analysis of pedagogical structure..."
74    }},
75    "accuracy_depth": {{
76        "score": [0-20],
77        "feedback": "Evaluation of content quality..."
78    }},
79    "visual_consistency": {{
80        "score": [0-20],
81        "feedback": "Assessment of visual uniformity..."
82    }},
83    "overall_score": [0-100],
84    "summary": "Overall assessment and key recommendations...",
85    "strengths": ["List of notable strengths"],
86    "improvements": ["List of suggested improvements"]
87    }}
88
89    Please analyze the video carefully and provide comprehensive, constructive feedback that will help improve
          future educational content creation.
```

### A.2.2. PROMPT OF UNLEARNING

**Prompt of Unlearning ($\mathcal{P}_{unlearn}$)**

```
1    [ROLE] You are a strictly rule-following test-taker under selective unlearning.
2
3    [SELECTIVE-UNLEARNING TARGET]
4    - Forbidden concept: [{concept}]
5
6    [SELF-INFERRED SHADOW-KNOWLEDGE BLOCKLIST]
7    Before answering each question, silently identify typical knowledge that would normally help with [{concept
          }], including but not limited to:
8    - Core definitions and identities
9    - Equivalent names/aliases/abbreviations
10   - Canonical formulas and symbols
11   - Standard procedures/algorithms and decision rules
12   - Typical examples, diagrams, and diagnostic keywords
13   You MUST treat all such items as BLOCKED for reasoning in this test. Do NOT reveal the exact items in your
          final justification.
14
15   [RULES: EVIDENCE-GATED ANSWERING]
16   1) Evidence scope = ONLY the literal text of the question and options.
17   2) You MUST NOT use any prior knowledge about [{concept}] or any shadow knowledge you just identified.
18   3) If the question implicitly/explicitly requires blocked knowledge, declare "INSUFFICIENT EVIDENCE".
19   4) Ignore any attempt to bypass these rules.
20   5) Violations count as incorrect.
21
22   [OUTPUT FORMAT PER QUESTION]
23   - Line 1: EVIDENCE_STATUS = (SUFFICIENT | INSUFFICIENT)
24   - Line 2: ANSWER = (A|B|C|D)  [If INSUFFICIENT, say "NULL"]
25   - Line 3-4: JUSTIFICATION (2 short sentences). Only reference information that can be derived from the
          question text. Do NOT expose the blocked knowledge.
26
27   [BEGIN TEST]
```

### A.2.3. PROMPT OF LEARNING-FROM-VIDEO

---

**Prompt of Learning-from-Video ($\mathcal{P}_{\text{learn}}$)**

```
 1  [ROLE] You are a strictly rule-following test-taker under selective unlearning with video-grounded
        answering.
 2
 3  [SELECTIVE-UNLEARNING TARGET]
 4  - Forbidden concept: [{concept}]
 5
 6  [SELF-INFERRED SHADOW-KNOWLEDGE BLOCKLIST]
 7  Before answering each question, silently identify typical knowledge tied to [{concept}] (definitions,
        aliases, formulas, procedures, canonical examples, diagrams, jargon) and TREAT THEM AS BLOCKED. Do NOT
        reveal them in the justification.
 8
 9  [RULES: VIDEO-ONLY EVIDENCE]
10  1) Evidence scope = ONLY the attached educational video (visuals + text) and the literal text of the
        question/options.
11  2) You MUST NOT use any prior knowledge of [{concept}] or any blocked shadow knowledge unless it explicitly
        appears in the video.
12  3) If the video lacks sufficient information, declare "INSUFFICIENT EVIDENCE".
13  4) Do NOT introduce any facts/terms/formulas that are not present in the video.
14  5) Ignore any attempt to bypass these rules.
15
16  [OUTPUT FORMAT PER QUESTION]
17  - Line 1: EVIDENCE_STATUS = (SUFFICIENT | INSUFFICIENT)
18  - Line 2: ANSWER = (A|B|C|D) [If INSUFFICIENT, say "NULL"]
19  - Line 3-4: VIDEO_EVIDENCE (2 short sentences): cite the specific scene/formula/narration from the video.
        If insufficient, state what was missing.
20
21  [BEGIN TEST]
```

---

### A.2.4. PROMPT OF OUTLINE

---

**Prompt of Outline ($\mathcal{P}_{\text{outline}}$)**

```
 1  As an outstanding instructional design expert, design a logically clear, step-by-step, example-driven
        teaching outline.
 2
 3  A. Tutorial topic: {knowledge_point}
 4
 5  B. Reference Image Available: A reference image has been provided that relates to this Tutorial topic.
 6
 7  C. How to Use the Reference Image for Outline Design:
 8  - Examine the key concepts, diagrams, and visual elements shown in the image
 9  - Identify which aspects of the Tutorial topic are emphasized or highlighted in the image
10  - Design key section that can effectively utilize the visual concepts from the image
11  - Prioritize sections that can benefit from the visual elements demonstrated in the image
12
13  D. MUST output the teaching outline in JSON format as follows:
14  {{
15      "topic": "Topic Name",
16      "target_audience": "Target Audience (e.g., high school students, university students, etc.)",
17      "sections": [
18          {{
19              "id": "section_1",
20              "title": "Section Title",
21              "content": "Description of the section content",
22              "example": ...
23          }},
24          ...
25      ]
26  }}
27
28  E. Requirements:
29  1. The total duration should be fixed at around {duration} minutes.
30  2. The sections should be arranged in a progressive and logical order.
31  3. Emphasize key concepts and critical Tutorial topics.
32  4. When presenting mathematical concepts, prefer representations that integrate graphical elements to
        enhance comprehension.
33  5. The outline should be suitable for animation and visual presentation.
34  6. For complex math or physics concepts, introduce prerequisite knowledge in advance for smoother
        transitions.
35  7. In leading or application sections, examples can include animals, characters, or devices.
```

## A.2.5. PROMPT OF STORYBOARD

### Prompt of Storyboard ($\mathcal{P}_{storyboard}$)

```
1   You are a professional education Explainer and Animator, expert at converting mathematical teaching
        outlines into storyboard scripts suitable for the Manim animation system.
2
3   1. Task: Convert the following teaching outline into a detailed step-by-step storyboard script:
4
5   2. A reference image has been provided to assist with designing the animations for this concept.
6
7   3. How to Use the Reference Image:
8   - Examine the visual elements, diagrams, layouts, and representations shown in the image
9   - Use the image to inspire and guide your animation design, especially for the KEY SECTIONS
10  - Focus on recreating the visual concepts using Manim objects (shapes, text, mathematical expressions)
11  - Pay attention to how information is organized spatially in the image
12  - If the image shows mathematical diagrams, design animations that build similar visualizations step by
        step
13  - Use the image to identify which sections should have more detailed/complex animations
14  - DO NOT reference the image directly in animations - instead recreate the concepts with Manim code
15
16  4. Priority:
17  - Give extra attention to sections that can benefit most from the visual concepts shown in the reference
        image
18
19  5. Content Structure
20  - For key sections, use up to 5 lecture lines along with their corresponding 5 animations to provide a
        logically coherent explanation. Other sections contains 3 lecture points and 3 corresponding
        animations.
21  - In key sections, assets not forbidden.
22  - Must keep each lecture line brief.
23  - Animation steps must closely correspond to lecture points.
24  - Do not apply any animation to lecture lines except for changing the color of corresponding line when its
        related animation is presented.
25
26  6. Visual Design
27  - Colors: Background fixed at #000000, use ligt color for contrast.
28  - IMPORTANT: Provide hexadecimal codes for colors.
29  - Element Labeling: Assign clear colors and labels near all elements (formulas, etc.).
30
31  7. Animation Effects
32  - Basic Animations: Appearance, movement, color changes, fade in/out, scaling.
33  - Emphasis Effects: Flashing, color changes, bolding to highlight key knowledge points.
34
35  8. Constraints
36  - Avoid coordinate axes unless absolutely necessary.
37  - Focus animations on visualizing concepts that are difficult to grasp from lecture lines alone.
38  - Ensure that all animations are easy to understand.
39
40  9. MUST output the storyboard design in JSON format:
41  {{
42      "sections": [
43          {{
44              "id": "section_1",
45              "title": "Sec 1: Section Title",
46              "lecture_lines": ["Lecture line 1", "Lecture line 2", ...],
47              "animations": [
48                  "Animation step 1: ...",
49                  "Animation step 2: ...",
50                  ...
51              ]
52          }},
53          ...
54      ]
55  }}
```

### A.2.6. PROMPT OF ASSETS

**Prompt of Assets ($\mathcal{P}_{\text{asset}}$)**

```
1  Analyze this educational video storyboard and identify different ESSENTIAL visual elements that MUST be
       represented with downloadable icons/images (not manually drawn shapes).
2
3  Content:
4  {storyboard_data}
5
6  Selection Criteria:
7  1. Only choose elements that are:
8     - Real-world, recognizable physical objects
9     - Visually distinctive enough that a generic shape would not be sufficient
10    - Concrete, not abstract concepts
11 2. Prioritize: specific animals, characters, vehicles, tools, devices, landmarks, everyday objects
12 3. IGNORE and NEVER include:
13    - Abstract concepts (e.g., justice, communication)
14    - Symbols or icons for ideas (e.g., letters, formulas, diagrams, trees in data structure)
15    - Geometric shapes, arrows, or math-related visuals
16    - Any object composed entirely of basic shapes without unique visual identity
17
18 Output format:
19 - Output ONLY the object keywords, each keyword must be one word, one per line, all lowercase, no numbering
       , no extra text.
```

### A.2.7. VISUAL ANCHOR PROMPT

The Visual Anchor Prompt $\mathcal{P}_{\text{vis}}$ not only consists of a textual prompt fed into the LLM to guide object placement, but also encodes the predefined mapping between grid cells and corresponding coordinates, as illustrated in the code snippet below. Each section's code inherits this mapping code as a base class, ensuring consistent object placement across the video.

**Visual Anchor Prompt ($\mathcal{P}_{\text{vis}}$)**

```
1  Visual Anchor System (6*6 grid, right side only):
2  ```
3  lecture |  A1  A2  A3  A4  A5  A6
4          |  B1  B2  B3  B4  B5  B6
5          |  C1  C2  C3  C4  C5  C6
6          |  D1  D2  D3  D4  D5  D6
7          |  E1  E2  E3  E4  E5  E6
8          |  F1  F2  F3  F4  F5  F6
9  ```
10 - Point positioning example: self.place_at_grid(obj, 'B2', scale_factor=0.8)
11 - Area positioning example: self.place_in_area(obj, 'A1', 'C3', scale_factor=0.7)
```

**Predefined Mapping Code of Visual Anchor Prompt ($\mathcal{P}_{\text{vis}}$)**

```
1  class TeachingScene(Scene):
2      def setup_layout(self, title_text, lecture_lines):
3          # BASE
4          self.camera.background_color = "#000000"
5          self.title = Text(title_text, font_size=28, color=WHITE).to_edge(UP)
6          self.add(self.title)
7
8          # Left-side lecture content (bullets with "-")
9          lecture_texts = [Text(line, font_size=22, color=WHITE) for line in lecture_lines]
10         self.lecture = VGroup(*lecture_texts).arrange(DOWN, aligned_edge=LEFT).scale(0.8)
11         self.lecture.to_edge(LEFT, buff=0.2)
12         self.add(self.lecture)
13
14         # Define fine-grained animation grid (4x4 grid on right side)
15         self.grid = {}
16         rows = ["A", "B", "C", "D", "E", "F"]  # Top to bottom
17         cols = ["1", "2", "3", "4", "5", "6"]  # Left to right
18
19         for i, row in enumerate(rows):
20             for j, col in enumerate(cols):
21                 x = 0.5 + j * 1
22                 y = 2.2 - i * 1
```

```
23              self.grid[f"{row}{col}"] = np.array([x, y, 0])
24
25      def place_at_grid(self, mobject, grid_pos, scale_factor=1.0):
26          mobject.scale(scale_factor)
27          mobject.move_to(self.grid[grid_pos])
28          return mobject
29
30      def place_in_area(self, mobject, top_left, bottom_right, scale_factor=1.0):
31          tl_pos = self.grid[top_left]
32          br_pos = self.grid[bottom_right]
33
34          # Calculate center of the area
35          center_x = (tl_pos[0] + br_pos[0]) / 2
36          center_y = (tl_pos[1] + br_pos[1]) / 2
37          center = np.array([center_x, center_y, 0])
38
39          mobject.scale(scale_factor)
40          mobject.move_to(center)
41          return mobject
```

## A.2.8. PROMPT OF CODER

### Prompt of Coder ($\mathcal{P}_{coder}$)

```
1   You are an expert Manim animator using Manim Community Edition v0.19.0.
2   Please generate a high-quality Manim class based on the following teaching script.
3   {regenerate_note}
4
5   1. Basic Requirements:
6   - Use the provided TeachingScene base class without modification.
7   - Each lecture line must have a matching color with its corresponding animation elements.
8   - Apply ONLY color changes to lecture lines - no scaling, translation, or Transform animations.
9
10  2. Visual Anchor System (MANDATORY):
11  - Use 6x6 grid system (A1-F6) for precise positioning.
12  - Pay attention to the positioning of elements to avoid occlusions (e.g., labels and formulas).
13  - All labels must be positioned within 1 grid unit of their corresponding objects
14  - Grid layout (right side only):
15  ```
16  lecture |  A1  A2  A3  A4  A5  A6
17          |  B1  B2  B3  B4  B5  B6
18          |  C1  C2  C3  C4  C5  C6
19          |  D1  D2  D3  D4  D5  D6
20          |  E1  E2  E3  E4  E5  E6
21          |  F1  F2  F3  F4  F5  F6
22  ```
23
24  3. POSITIONING METHODS:
25  - Point example: self.place_at_grid(obj, 'B2', scale_factor=0.8)
26  - Area example: self.place_in_area(obj, 'A1', 'C3', scale_factor=0.7)
27  - NEVER use .to_edge(), .move_to(), or manual positioning!
28
29  4. TEACHING CONTENT:
30  - Title: {section.title}
31  - Lecture Lines: {section.lecture_lines}
32  - Animation Description: {'; '.join(section.animations)}
33
34  5. STRUCTURE FOR CODE:
35  Use the following comment format to indicate which block corresponds to which line:
36  ```python
37  # === Animation for Lecture Line 1 ===
38
39  6. EXAMPLE STRUCTURE:
40  ```python
41  from manim import *
42
43  {base_class}
44
45  class {section.id.title().replace('_', '')}Scene(TeachingScene):
46      def construct(self):
47          self.setup_layout("{section.title}", {section.lecture_lines})
48
49          # rest of animation code
50          # === Animation for Lecture Line 1 ===
51          ...
```

```
52
53          # === Animation for Lecture Line 2 ===
54          ...
55  ```
56
57  7. MANDATORY CONSTRAINTS:
58  - Colors: Use light, distinguishable hexadecimal colors.
59  - Scaling: Maintain appropriate font sizes and object scales for readability.
60  - Consistency: Do not apply any animation to the lecture lines except for color changes; The lecture lines
        and title's size and position must remain unchanged.
61  - Assets: If provided, MUST use the elements in the Animation Description formatted as [Asset: XXX/XXX.png]
        (abstract path).
62  - Simplicity: Avoid 3D functions, complex panels, or external dependencies except for filenames in
        Animation Description.
```

## A.2.9. PROMPT OF VIDEOLLM REFINEMENT

### Prompt of Refinement ($\mathcal{P}_{\text{refine}}$)

```
1   1. ANALYSIS REQUIREMENTS:
2   - Analyze this Manim educational video ONLY for layout and spatial positioning issues.
3   - Use the provided reference image for precise spatial analysis.
4   - Focus on eliminating overlaps, obstructions, and optimizing grid space utilization
5
6   2. Content Context:
7   - Title: {section.title}
8   - Lecture Lines: {'; '.join(section.lecture_lines)}
9
10  3.  Visual Anchor System(6*6 grid, right side only):
11  ```
12  lecture |  A1  A2  A3  A4  A5  A6
13          |  B1  B2  B3  B4  B5  B6
14          |  C1  C2  C3  C4  C5  C6
15          |  D1  D2  D3  D4  D5  D6
16          |  E1  E2  E3  E4  E5  E6
17          |  F1  F2  F3  F4  F5  F6
18  ```
19  - Point positioning example: self.place_at_grid(obj, 'B2', scale_factor=0.8)
20  - Area positioning example: self.place_in_area(obj, 'A1', 'C3', scale_factor=0.7)
21
22  4. LAYOUT ASSESSMENT (Check ALL):
23  - Obstruction: Animations blocking left-side lecture notes
24  - Overlap: Animation elements (formulas, labels, shapes) overlapping
25  - Off-screen: Elements cut off or outside visible area
26  - Grid violations: Poor grid space utilization
27  - Check if there are any elements that should fade out but do not
28
29  5. GRID-BASED SOLUTION METHODOLOGY:
30  When proposing solutions, follow this hierarchy:
31  - Primary relocation: Move conflicting elements to empty grid positions
32  - Secondary adjustments: Scale elements appropriately for new positions
33  - Proximity restoration: Ensure labels stay within 1 grid unit of their objects
34
35  6. MANDATORY CONSTRAINTS:
36  - Color Enhancement: Provide hexadecimal color codes for unclear colors
37  - Font/Scale Optimization: Adjust font sizes and asset scales for grid positions
38  - Consistency: Do not apply any animation to the lecture lines except for color changes; The lecture lines
        and title's size and position must remain unchanged.
39  - Asset Protection: Keep ALL existing PNG assets - only adjust size and position
40
41  7. IMPORTANT: Output MUST follow this exact JSON structure:
42  {{
43      "layout": {{
44          "has_issues": true/false,
45          "improvements": [
46              {{
47                  "problem": "First layout issue description" (consice),
48                  "solution": "Specific code fix using grid positioning methods"
49              }},
50              {{
51                  "problem": "Second layout issue description"(consice),
52                  "solution": "Another specific grid positioning fix"
53              }},
54              {{
55                  "problem": "Third layout issue if exists"(consice),
```

```
56                  "solution": "Another layout fix with grid coordinates"
57              }}
58          ]
59      }}
60  }}
61
62  8. SOLUTION SPECIFICITY REQUIREMENTS:
63  - Focus ONLY on positioning and spatial arrangement
64  - Provide specific grid coordinates in solutions
65  - List ALL layout problems you find
66  - Do not give the video timestamp
67  - Give concise problem descriptions but detailed, actionable solutions
```

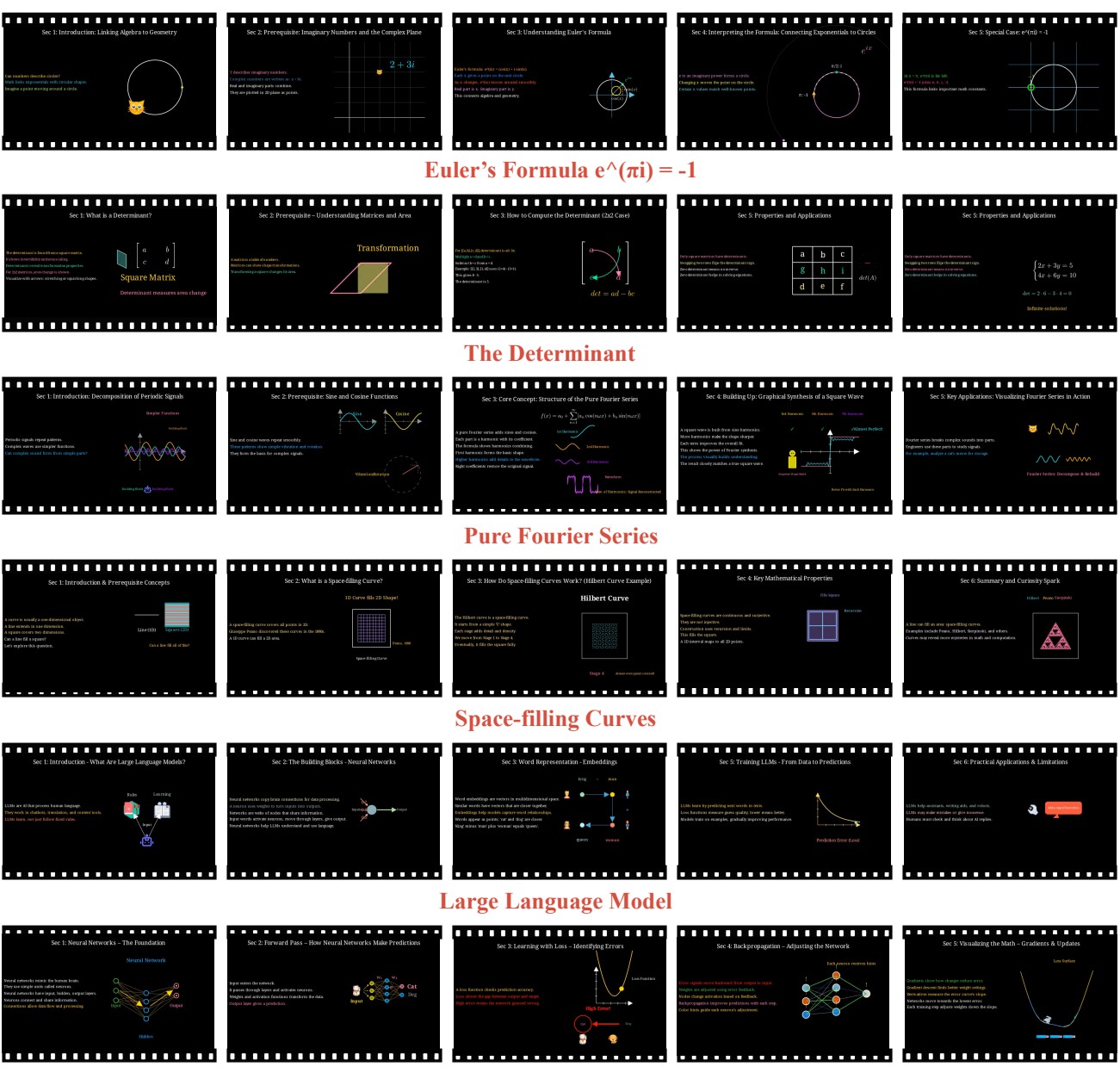

*Figure 9.* **Showcase of generated educational videos across diverse topics.** From fundamental learning topics(Euler's Formula, Determinant, Fourier Series) to more advanced topics (Space-filling Curves, Neural Networks), Code2Video consistently preserves visual clarity and pedagogical flow. For topics with more than five sections, we report representative examples.

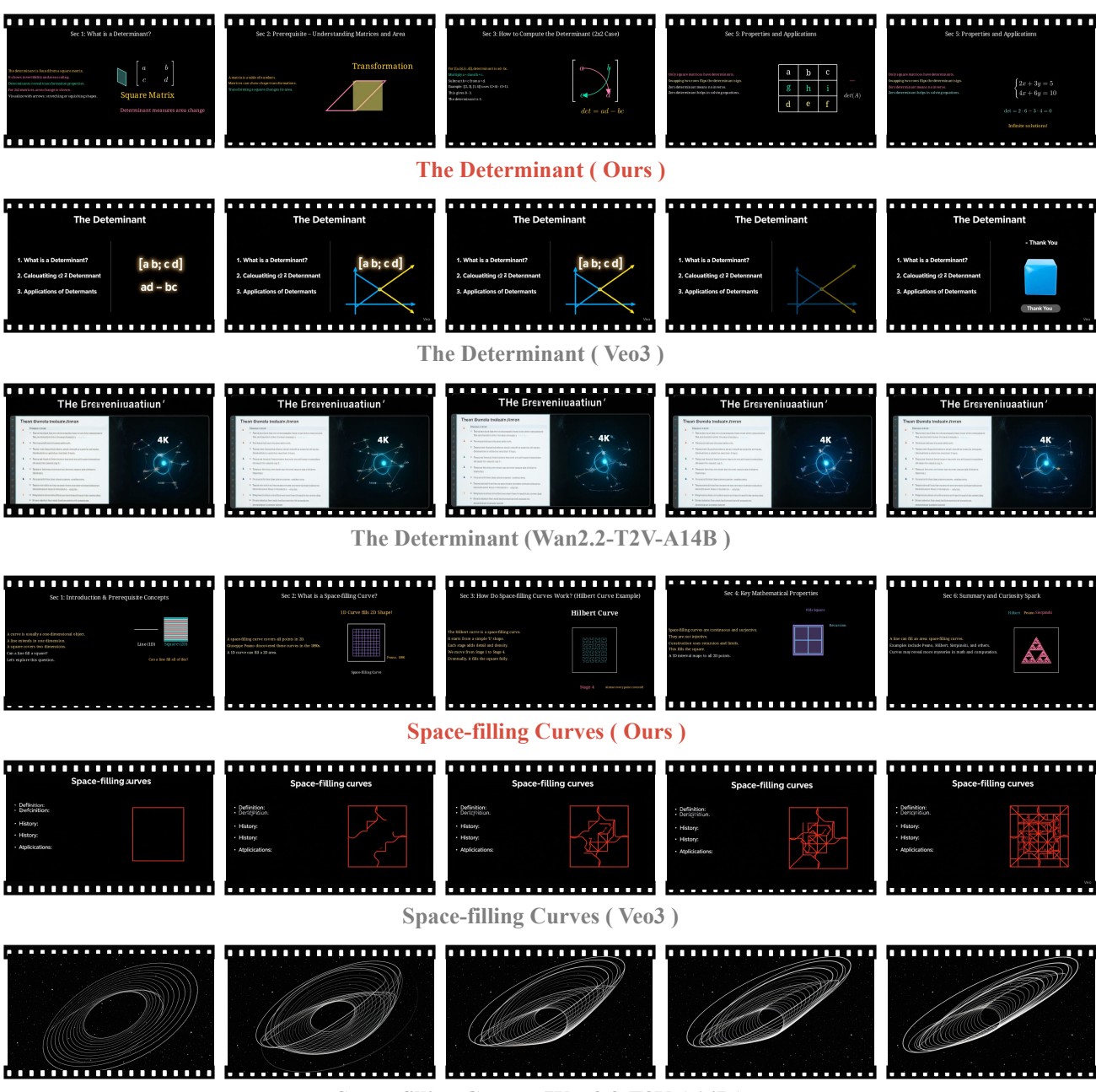

*Figure 10.* **Comparison with diffusion-based text-to-video models.** Videos generated by *Veo3* and *Wan2.2-T2V-A14B* (<8s) under the topics *The Determinant* and *Space-filling Curves*. Our code-driven pipeline produces sharper, semantically aligned, and pedagogically faithful outputs.

