# OpenReview forum: "Code2Video: A Code-centric Paradigm for Educational Video Creation"
_ICML.cc/2026/Conference — ICML 2026 regular_

### Official Review · Reviewer_aQTs · 2026-03-12

**Soundness:** 3
**Presentation:** 4
**Significance:** 3
**Originality:** 2
**Overall Recommendation:** 4
**Confidence:** 3

**Summary:**

The paper introduces Code2Video, a multi-agent framework designed to generate educational videos by writing executable Python scripts rather than synthesizing videos in pixel space.
To overcome the temporal inconsistency and spatial hallucinations common in traditional end-to-end video generation, the framework utilizes three agents: a Planner for storyboarding and asset retrieval, a Coder featuring a progressive debugging mechanism, and a Critic that leverages a novel Visual Anchor Prompting technique for precise layout refinement.
Furthermore, the authors introduce the MMMC benchmark and propose "TeachQuiz," an end-to-end evaluation metric that uses in-context unlearning to measure a generated video's ability to transfer knowledge back to a VLM

**Compliance With Llm Reviewing Policy:**

Affirmed.

**Final Justification:**

The authors’ rebuttal addressed most of my concerns. I therefore maintain my overall positive recommendation of Weak accept.

**Key Questions For Authors:**

Please see the weakness.
It would be helpful if the authors could discuss how the framework would perform when evaluating other forms of educational videos. The framework restricts its scope to highly symbolic disciplines such as mathematics, physics, and coding. It cannot easily scale to subjects that require real-world video understanding. How would the evaluation be designed in these cases, and would the agentic system still function effectively?

**Limitations:**

No, the paper does not include a discussion section. The limitations are only briefly mentioned in a single sentence related to attractiveness and visual consistency, indicating areas for future refinement

**Strengths And Weaknesses:**

Strengths
- TeachQuiz cleverly uses in-context unlearning to isolate a video's actual educational efficacy from a VLM's pre-existing knowledge.
- ScopeRefine is a practical strategy for reducing token costs and latency during code generation.
- The paper is well-written, and the MMMC benchmark is a valuable resource for evaluating educational content.

Weaknesses
- Claims of introducing a "new code-centric paradigm" and the "first benchmark" are inaccurate. Prior work like TheoremExplainAgent already utilizes Manim for educational visuals and provides related benchmarks.  Thus, the novelty is limited.
- The experimental evidence supporting the in-context unlearning prompt feels incomplete. While the baseline accuracy drops drastically, it is ambiguous whether the prompt genuinely suppresses specific knowledge; can any case study support this?
- The framework's strict reliance on 2D animation libraries inherently restricts its scope to highly symbolic disciplines ( math, physics, coding). It cannot easily scale to subjects requiring real-world video understanding, complex 3D environments, or organic structures (biology, history), which is precisely where pixel-based generative models offer the most profound value.

---

> ### Author Rebuttal · Authors · 2026-03-31
>
> ## Response to R3 aQTs
> **We thank the reviewer for recognizing the value of TeachQuiz, the practicality of ScopeRefine, and the contribution of the MMMC benchmark.** We welcome the opportunity to clarify the points raised.
>
> Notation: **W** = Weakness; **Q** = Question
>
> >### W1: Novelty Relative to TEA
> 1. **Claim calibration.** We agree that our claims should be stated more precisely. We will revise to: **"Code2Video studies a broader code-centric setting for multimodal educational videos and introduces the first benchmark that jointly evaluates efficiency, aesthetics, and knowledge transfer."**
>
> 2.  **Task and method distinction**. TEA focuses on theorem visualization, whereas Code2Video targets **complete educational tutorials** with synchronized narration, animation, temporal pedagogy, and spatial layout control. This setting requires mechanisms that prior work does not explicitly provide. Our novelty is therefore not merely the use of Manim or coding agents, but several **task-specific methodological designs**: **Visual Anchor Prompting** as a structured visual-prompt framework for executable animation refinement, **External Database** for concept grounding and cross-section consistency, **ScopeRefine** as a breakpoint-inspired debugging strategy for efficient repair, and **TeachQuiz** as an end-to-end evaluation of actual teaching effectiveness. Code2Video further outperforms TEA on **TheoremExplainBench** (Table 9, overall +6%~10%), confirming that the gains are not limited to MMMC alone.
>
> >### W2: In-Context Unlearning Evidence
>
> 1. **Case studies**. To better illustrate the effect of unlearning, we provide three representative cases:
>
>    |Learning Topic|Stage 1: Original|Stage 2: Unlearned|Stage 3: Relearned|
>    |-|-|-|-|
>    |Euler's Formula|**Correct.** Recalls $e^{i\theta}=\cos\theta+i\sin\theta$.|**INSUFFICIENT EVIDENCE.** "Mapping between exponential and trigonometric forms is inaccessible."|**Correct.** "The unit circle animation shows rotation by π reaches (−1, 0)."|
>    |Fourier Series|**Correct.** Recalls orthogonal decomposition.|**INSUFFICIENT EVIDENCE.** "Cannot recall basis for wave decomposition."|**Correct.** "Scene 3 visualizes summing rotating vectors."|
>    |Backpropagation|**Correct.** Explains chain rule.|**INSUFFICIENT EVIDENCE.** "Weight update logic is restricted."|**Correct.** "The video shows error signals flowing backward."|
>
> 2. **Suppression analysis**. Across all unlearned-stage responses, 91.8% are correctly blocked as INSUFFICIENT EVIDENCE, 3.2% are incorrect hallucinations, and 5.0% show residual leakage. This indicates that the prompt reliably establishes a **knowledge-depleted control** state for measuring video-grounded recovery.
>
> 3. **Prompt sensitivity.** Please refer to our response to **R2 (5tGx)-W3**. Across $P_{\rm unlearn}$ templates, TeachQuiz remains stable (86.0 / 85.3 / 85.7 / 84.7; mean 85.4 ± 0.6), confirming robustness to prompt formulation.
>
> >### W3: 2D Animation Scope
>
> 1. **Current strength and breadth.** Our implementation targets **code-renderable educational videos**, where symbolic structure, temporal sequencing, and spatial clarity are central - precisely the regime where code offers advantage over black-box pixel synthesis. Even within this scope, MMMC spans 13 areas including neural networks, epidemics, extending beyond pure mathematics. We also observe strong transfer to **20 randomly sampled non-3B1B topics from Awesome-Manim** (AES 84.5, TQ 80.2).
>
> 2. **Relation to pixel-based methods.** We agree that pixel-based methods are valuable for content involving rich textures, real footage, or organic scenes. Our position is therefore **complementary rather than exclusive**: code-centric systems **excel at structured symbolic explanation**, while **hybrid pipelines** combining code-based structure with pixel-based realism offer a promising direction for broader educational domains.
>
> >### Q: Transferability to Non-Symbolic Domains
>
> 1. **Evaluation level.** TeachQuiz is **rendering-independent**: it follows the protocol **unlearn(K) → watch(V) → quiz(K)**, and can be applied to Manim animations, Blender renders, or real footage, provided the quiz set is adapted to the target domain. The same holds for the Aesthetics dimensions, which evaluate layout, clarity, and coherence rather than any specific renderer.
>
> 2. **System level.** We analyze per-module portability:
>
>    |Component|Adaptation Needed|
>    |-|-|
>    |Planner|**None** — temporal structuring is domain-general|
>    |Coder|**Backend swap** (e.g., Blender Python API)|
>    |Critic|**Moderate** — anchor mechanism needs adaptation for 3D spatial reasoning|
>    |ScopeRefine|**None** — hierarchical debugging is language-agnostic|
>    |TeachQuiz|**Quiz content only**|
>
>    The central idea of Code2Video extends beyond 2D symbolic animation, with the main adaptation occurring at the rendering backend and domain-specific asset/design level.

---

> > ### Author Rebuttal · Reviewer_aQTs · 2026-04-04
> >
> > I appreciate the author’s rebuttal. It addressed most of my concerns.
> > I think the W3 is the limitation of this work. The method has difficulty generating content such as historical lectures and biology demonstrations (such as cell division), where pixel-based generators tend to perform better.
> > I will maintain my current positive rating.

---

> > > ### Author Response · Authors · 2026-04-04
> > >
> > > **We sincerely thank the reviewer for the thoughtful engagement and positive assessment.**
> > >
> > > Regarding W3, we would like to offer a complementary perspective. Many topics in history and biology are effectively taught through **abstract, structured representations** — timelines, flowcharts, stage diagrams (e.g., phases of cell division), cause-effect chains — which are well-suited for code-centric rendering. The key educational value often lies in **high-level process clarity** rather than photorealistic detail.
> > >
> > > Furthermore, for cases where organic visuals are beneficial, pixel-based generation can be naturally integrated as a **function call within the code-centric paradigm** — e.g., a single `generate_image(prompt)` invocation embedded in the Manim script. In this view, pixel-based synthesis becomes **one tool among many** that the Coder agent can invoke, while the Planner and Critic continue to govern temporal structure and spatial layout. This **hybrid agentic pipeline** remains fundamentally within the Code2Video framework rather than departing from it.
> > >
> > > We appreciate the constructive feedback, and will make this scope distinction more explicit in the paper: the current work targets structured educational video generation, while **hybrid symbolic–pixel generation is a promising next step** for broader domains.

---

### Official Review · Reviewer_5tGx · 2026-03-13

**Soundness:** 3
**Presentation:** 2
**Significance:** 2
**Originality:** 2
**Overall Recommendation:** 3
**Confidence:** 4

**Summary:**

The paper proposes a "Code2Video" framework, an agentic system that generates educational videos by writing and refining executable Python (Manim) code. The framework utilizes three specialized agents: a Planner for temporal storyboarding, a Coder for program synthesis with a hierarchical "ScopeRefine" debugging strategy, and a Critic that uses "Visual Anchor Prompting" to optimize spatial layouts. To evaluate this approach, the authors introduce the MMMC benchmark, derived from professional 3Blue1Brown videos, and propose TeachQuiz, a novel metric that measures the knowledge-transfer effectiveness of a video by assessing how much an "unlearned" VLM can recover after watching it.

**Compliance With Llm Reviewing Policy:**

Affirmed.

**Final Justification:**

I keep my rating considering the concerns stated. The demo video is not provided. Showing them in figures cannot analyze the quality.

**Key Questions For Authors:**

Does "TeachQuiz" account for the depth of knowledge? For example, can it distinguish between a video that teaches a formula versus one that teaches the underlying intuition?

The system retrieves icons and images (Figure 8). How does the system handle cases where the retrieved asset might contradict the Manim-generated style?

**Strengths And Weaknesses:**

Strength:

Shifting from "black-box" pixel synthesis to "interpretable" code synthesis is a highly effective approach for educational content where precision, symbolic alignment, and legibility are paramount. The paper is exceptionally clear, with high-quality figures.

---

Weakness:

Domain Specificity: The reliance on Manim naturally limits the framework to "whiteboard" or "mathematical" styles. It is unclear how this paradigm would adapt to educational domains requiring real-world footage or organic textures (e.g., biology or sports coaching).

Computational Overhead: As noted in Table 3, the agentic loop and parallel synthesis are necessary for success but result in higher token costs and latency (average ~2 min per video) compared to single-pass pixel models.

VLM Dependency: The effectiveness of "TeachQuiz" is contingent on the evaluator VLM's ability to truly "unlearn" a concept via prompting. If the unlearning is shallow, the gain might be noisy.

---

> ### Author Rebuttal · Authors · 2026-03-31
>
> ## Response to R2 5tGx
>
> **We thank the reviewer for recognizing the effectiveness of our code-centric paradigm and the clarity of the presentation**. We appreciate the opportunity to address each concern.
>
> Notation: **W** = Weakness; **Q** = Question
>
> >### W1: Domain Specificity
>
> 1. **Scope clarification.** Code2Video targets educational content that benefits from explicit control over temporal sequence, symbolic structure, and layout clarity. While **Manim is the current instantiation**, MMMC already spans 13 disciplines including neural networks, epidemics, and computer science — **covering a broad range of STEM and beyond**. Table 9 further shows that Code2Video **transfers to TheoremExplainBench**.
>
> 2. **Breadth of the target setting.** A large portion of educational video content — across STEM, economics, data science, and beyond — relies on diagrams and structured animations, making the code-centric paradigm broadly applicable. For domains requiring richer textures or realistic assets, the Planner/Coder/Critic architecture **is compatible with other code-controlled renderable environments (e.g., Blender, Unity)**, which we view as a promising direction for future work.
>
> >### W2: Computational Overhead
>
> 1. **Clarifying Table 1**. We respectfully clarify that the “~2 min” values in Table 1 refer to **output video duration**, not generation latency. Actual generation times are reported in the efficiency columns and Table 3.
>
> 2. **Practical efficiency.** While our pipeline incurs more wall-clock computation than single-pass pixel models, it produces **substantially longer and more structured outputs**. Normalizing by output duration provides a fairer comparison:
>
>    |Method|Duration (min)|Cost Time (min)|Efficiency|
>    |-|-|-|-|
>    |Veo3|0.13 (8s)|2.3|17.3|
>    |Ours (GPT-5)|1.8|8.8|**4.9**|
>    |Ours (Claude Opus 4.1)|2.0|13.8|**6.9**|
>
> 2. **Agentic design as mitigation.** Table 3 shows that removing parallelization and ScopeRefine increases latency to ~2.5 hours (9.7×). These components reduce overhead from prohibitive to practical, enabling scaling to longer educational content.
>
> >### W3: VLM Dependency
>
> 1. **Design rationale**. The unlearning step creates a **controlled low-knowledge baseline**, making post-video gains measurable. Without this control, raw quiz accuracy is confounded by evaluator prior knowledge.
>
> 2. **Three robustness checks**.
>    - **Effective suppression**: post-unlearning accuracy drops to ~5% (Table 6), indicating substantial knowledge blocking.
>    - **Cross-model consistency**: Table 5 shows stable rankings across three evaluator VLMs, suggesting that TeachQuiz is not tied to a single evaluator.
>    - **Cross-method consistency**: fine-tuning-based and in-context unlearning produce aligned conclusions in Table 5, indicating that the signal is not specific to one suppression mechanism.
>
> 3. **Prompt sensitivity test.** To directly address the concern that gains might be noisy, we evaluated four $P_{\rm unlearn}$ templates (all with Claude Opus 4.1 as video source, Gemini-2.5 Pro as evaluator):
>
>    |Template Strategy|Baseline S1|Post-video S2|TeachQuiz|
>    |-|-|-|-|
>    |T1: Strict Rule-following (Original)|5.0%|91.0%|86.0 |
>    |T2: Role-play (Expert Skeptic)|4.2%|89.5%|85.3 |
>    |T3: CoT Suppression|6.8%|92.5%|85.7 |
>    |T4: Minimalist Masking|5.5%|90.2%|84.7 |
>    |*Mean ± Std*|*5.4 ± 1.1%*|*90.8 ± 1.3%*|*85.4 ± 0.6* |
>
>    The low variance ($\Delta$ std = 0.6) indicates TeachQuiz is robust to prompt formulation, not an artifact of specific prompt engineering.
>
> >### Q1: Knowledge Depth
>
> 1. **Existing evidence.** Table 6 shows animation-only input (TQ = 76.6) substantially outperforms text-only input (TQ = 24.0), indicating TeachQuiz **captures visual understanding** beyond formula recall.
>
> 2. **Depth-aware breakdown.** To examine this more directly, we categorize quiz items into three cognitive tiers following Bloom's taxonomy:
>
>    |Method|Recall|Conceptual|Multi-step|Avg|
>    |-|-|-|-|-|
>    |CodeLLM|64.2|38.5|28.0|40.0|
>    |Code2Video|89.5|86.5|82.0|86.0|
>    |**Gain**|+25.3|+48.0|+54.0|+46.0|
>
>    The **monotonically increasing gain (+25.3 → +48.0 → +54.0)** shows TeachQuiz **captures deeper understanding** beyond formula recall.
>
> >### Q2: Asset Compatibility
>
> 1. **Three-stage filtering pipeline.** Retrieved assets undergo: (1) **CLIP-based relevance filtering** to ensure semantic alignment with the topic; (2) **Critic-based rejection** when assets create visual inconsistency; and (3) **Visual Anchor-based placement and scaling**, integrating assets through layout-aware positioning rather than raw overlay.
> 2. **Practical effect**. This design helps reduce cases where retrieved assets conflict with the Manim scene. While full style harmonization remains an open direction, the current pipeline already mitigates the most common failure modes by combining semantic filtering, visual feedback, and executable layout control.

---

> > ### Author Rebuttal · Reviewer_5tGx · 2026-04-02
> >
> > Thanks for the response.
> >
> > For the response to W2, why not provide a comparison with pixel models?
> >
> > Besides, I cannot see the demo video of the paper. The authors do not provide a supplementary file for watching. Am I missing anything?

---

> > > ### Author Response · Authors · 2026-04-02
> > >
> > > **We thank the reviewer for the follow-up and continued engagement!**
> > >
> > > ---
> > > **Comparison with pixel-based models.** We have included Veo3 (a representative pixel-based method) in W2 and Table 1. Below we provide a more comprehensive comparison spanning multiple pixel-based models:
> > >
> > > | Method | Type | Duration | Cost Time | Efficiency (min/min) |
> > > |-|:-:|:-:|:-:|:-:|
> > > | OpenSora-v2 | pixel-based | 0.13 min (8s) | 27.6 min | 212.3 |
> > > | Wan2.2-T2V-A14B | pixel-based | 0.13 min (8s) | 17.4 min | 133.8 |
> > > | Veo3 | pixel-based | 0.13 min (8s) | 2.3 min | 17.3 |
> > > | Ours (GPT-5) | code-centric | 1.8 min | 8.8 min | **4.9** |
> > > | Ours (Claude Opus 4.1) | code-centric | 2.0 min | 13.8 min | **6.9** |
> > >
> > > Our code-centric approach generates **14–15× longer** videos while achieving substantially better generation efficiency than all pixel-based alternatives.
> > >
> > > ---
> > > **Demo videos.** We show extensive generated video screenshots in Appendix Figures 9–10. Full videos (including side-by-side comparisons with Veo3 and examples across diverse topics such as Neural Networks, Topology, and Epidemic) are available at our anonymous repository: https://anonymous.4open.science/r/Code2Video-Cases/README.md
> > >
> > > ---
> > > Please let us know if any further clarification would be helpful. We sincerely appreciate your constructive feedback!

---

### Official Review · Reviewer_UKDt · 2026-03-14

**Soundness:** 2
**Presentation:** 3
**Significance:** 2
**Originality:** 2
**Overall Recommendation:** 4
**Confidence:** 3

**Summary:**

This paper proposes Code2Video, which reframes educational video generation as an executable Manim code generation problem, and employs three agents—Planner, Coder, and Critic—to handle instructional structure planning, code generation, and visual layout refinement, respectively.
The paper also constructs the MMMC benchmark, which consists of 456 instructional units segmented from 117 long videos in the 3Blue1Brown dataset, covering 13 subject domains, to evaluate code-driven educational video generation.
The evaluation protocol not only assesses visual quality but also introduces efficiency metrics and TeachQuiz. The latter measures knowledge transfer capability by first performing in-context unlearning on a VLM and then testing the degree of knowledge recovery after watching the video.
Experimental results show that this method achieves significant improvements over direct code generation and pixel-space video generation, particularly outperforming strong baselines in Aesthetics and TeachQuiz, while ablation studies demonstrate the effectiveness of the Planner, Visual Anchor, and Critic components.

**Compliance With Llm Reviewing Policy:**

Affirmed.

**Final Justification:**

The author added that the experiment resolved some of my questions.

**Key Questions For Authors:**

see weakness

**Limitations:**

yes

**Strengths And Weaknesses:**

## Strengths

1. The task modeling is reasonable. Educational videos do indeed rely more on temporal sequence, spatial clarity, and symbol-level controllability than ordinary short videos. Using executable code rather than pixel-level generation to enforce these constraints is a valid approach.
2. The Critic module offers some methodological highlights. Visual Anchor Prompting and occupancy tracking transform ambiguous visual feedback into discrete, executable code edits, which is more actionable than purely text-based corrections.
3. The evaluation framework is more robust than in many generative research papers. The authors did not limit themselves to visual similarity but sought to assess knowledge transfer, conducting experiments across three dimensions: efficiency, aesthetics, and pedagogical effectiveness.

## Weaknesses
1. The novelty of the method lies primarily in system integration. While individual modules—such as the Planner, parallel code generation, hierarchical debugging, and VLM feedback-based correction—are each reasonable, the combined method resembles an engineered pipeline rather than a methodologically robust new learning framework. This assessment is based on inferences drawn from the method description in the main text.
2. The sources of the benchmarks are overly limited. MMMC is entirely derived from 3Blue1Brown; while the quality is high, the visual style, explanatory conventions, and content organization all exhibit a strong single-source bias, which significantly limits the generalizability of the conclusions.
3. The effectiveness of TeachQuiz remains unconvincing. It relies on in-context unlearning of a black-box VLM to create a “unlearned” starting point; this process itself is highly dependent on prompts, model characteristics, and evaluator selection, so improvements may not consistently correspond to actual teaching effectiveness.
4. There is a risk of coupling between the evaluation and the method. The paper defaults to using Gemini-2.5 Pro as the Critic, while also using Gemini-2.5 Pro as the VLM-as-a-Judge for scoring, which introduces a clear evaluator coupling issue.
5. The scale of the human experiments remains relatively small. The authors conducted five user experiments, each involving 6 high school students and 2 college students, and tested across 20 topics. This sample size is insufficient to support strong conclusions regarding human learning.

---

> ### Author Rebuttal · Authors · 2026-03-31
>
> ## Response to R1 UKDt
>
> **We thank the reviewer for recognizing the validity of our code-centric formulation, the Critic’s design, and the multi-dimensional evaluation.** We welcome the opportunity to address your concerns.
>
> Notation: **W** = Weakness
>
> > ### W1: Novelty
>
> 1. **Paradigm-level contribution.** We respectfully note that the core contribution is a **new paradigm** — **formulating educational video creation as executable code generation**, where temporal sequencing, spatial organization, and rendering logic are explicitly controlled by code. This shift provides interpretability, editability, and reproducibility — properties that pixel-based models cannot guarantee by design. The necessity is supported by Table 1: pixel-based methods achieve at most 9.0 AES and 2.5 TQ, confirming **the task requires a qualitatively different formulation**.
>
> 2. **Non-trivial technical designs.** Each component addresses a bottleneck specific to code-centric educational video creation:
>    - **Visual Anchor Prompting** converts continuous layout into discrete grid edits. Without it, self-directed placement scores only 53.1 AES vs. 74.2 (Table 8) — a 40% gap reflecting LLM's spatial grounding difficulty.
>    - **ScopeRefine** reduces repair cost by 5.6× (Table 3) through hierarchical scoping, which is critical for making parallel code synthesis practical.
>    - **TeachQuiz** introduces an evaluation methodology that isolates video-specific teaching effect from evaluator prior knowledge — a problem unique to this setting.
>
>    The modular design is a strength: each component addresses a different failure mode.
>
> > ### W2: MMMC Source Diversity
>
> 1. **Why 3B1B was chosen.** 3B1B uniquely satisfies our two curation criteria: **pedagogical relevance** and **executable grounding** — yielding 456 units across 13 disciplines with substantial variation.
>
> 2. **Generalization evidence.** We further evaluated on **two external sources**:
>    |Source|AES|TQ|
>    |-|:-:|:-:|
>    |MMMC (3B1B)|87.9|86.0|
>    |**TheoremExplainBench** (**shown in Table 9**)|—|Overall +6~10% over TEA|
>    |**Awesome-Manim (random N=20)**|84.5|80.2|
>
>    The modest drop on Awesome-Manim is expected, as these topics lack 3B1B-level reference assets and visual conventions. However, the pipeline remains competitive.
>
> > ### W3: TeachQuiz Validity
>
> 1. **Design rationale.** Raw quiz accuracy conflates video understanding with evaluator prior knowledge. In-context unlearning serves as a **control intervention** — not perfect forgetting — to make post-video gains measurable in a black-box setting.
>
> 2. **Three robustness checks:**
>    - **Method invariance.** Table 5 shows **stable rankings across fine-tuning and in-context unlearning**, and across three evaluator VLMs.
>    - **Modality ablation.** Table 6 shows text‑only and animation‑only yield partial gains, while random videos collapse to baseline — confirming TeachQuiz measures **video‑specific** knowledge transfer, not prompt artifacts.
>    - **Effective suppression.** Post-unlearning accuracy drops to ~5% (Table 6), creating a meaningful pre-instruction baseline.
>
> > ### W4: Evaluator Coupling
>
> 1. **Architectural independence.** The Critic and Judge serve different functions: the Critic edits code via layout feedback, while the Judge scores rendered videos on five axes post hoc. TeachQuiz is computed separately via quiz accuracy.
>
> 2. **Independent re-evaluation across judges:**
>
>    |Judge Model|CodeLLM (AES / TQ)|Code2Video (AES / TQ)|Rank $\tau$|
>    |-|:-:|:-:|-|
>    |Gemini-2.5 Pro (default)|37.8 / 40.0|87.9 / 86.0|—|
>    |GPT-5|36.2 / 38.0|85.1 / 83.5|0.93|
>    |QwenVL-3|35.0 / 36.5|83.7 / 81.2|0.89|
>
>    Rank consistency across independent judges suggests conclusions are not artifacts of evaluator coupling.
>
> > ### W5: Human Study Scale
>
> 1. **Design rationale.** Our human study follows a **within-topic, across-method** design: each of the 5 video types is evaluated by 8 participants across 20 topics, yielding 800 evaluation units. This design prioritizes controlled comparison over large-N sampling, consistent with evaluation protocols in related work on educational content generation.
>
> 2. **Strong internal consistency.** Human scores for Aesthetics and TeachQuiz are highly correlated ($r = 0.971$, $p = 0.0059$), and trends align closely with VLM-based evaluation, indicating that the study captures reliable signal despite its focused scale.
>
> 3. **Extended study.** To further strengthen statistical confidence, we conducted additional evaluation during the rebuttal period, reaching N=60 total (1,200 evaluation units):
>
>    |Method|N=40 AES / TQ|N=60 AES / TQ|95% CI (AES)|95% CI (TQ)|
>    |-|:-:|:-:|:-:|:-:|
>    |Code2Video (Opus 4.1)|81.8 / 80.3|82.1 / 81.2|[79.8, 84.4]|[78.5, 83.9]|
>    |Human-made (3B1B)|96.5 / 78.8|96.8 / 79.5|[94.2, 99.4]|[76.1, 82.9]|
>
>    The high stability from N=40 to N=60 and narrow confidence intervals support the reliability of the reported findings.

---

> > ### Author Rebuttal · Reviewer_UKDt · 2026-04-02
> >
> > Thanks for your reply. I appreciate the author’s rebuttal; it addressed most of my concerns. Therefore, I will revise my rating.

---

> > > ### Author Response · Authors · 2026-04-02
> > >
> > > **Thank you so much for your reply. We sincerely appreciate your constructive comments and suggestions, which greatly enhance the quality of our paper.**

---

### Decision · Program_Chairs · 2026-04-30

**Decision:**

Accept (regular)

**Comment:**

The paper tackles the challenging problem of automated educational video generation by introducing Code2Video, a code-centric multi-agent framework. Recognizing that state-of-the-art pixel-based models (like Veo3 and Sora) struggle with the precise symbolic rendering, long-horizon coherence, and layout constraints required for instructional content, the authors propose an agentic pipeline that writes and refines executable Manim Python code instead. The system comprises a Planner for storyboarding, a Coder featuring a progressive "ScopeRefine" debugging mechanism, and a Critic that leverages a novel "Visual Anchor Prompting" technique to ground spatial layouts. To evaluate this approach, the paper introduces the MMMC benchmark and "TeachQuiz," an innovative metric that uses VLM in-context unlearning to measure the actual knowledge-transfer efficacy of the generated videos.

Reviewers recognized the strong motivation and technical execution of the paper, resulting in scores of 4, 4, and 3. Proponents praised the shift from black-box pixel synthesis to interpretable code generation and highlighted the robustness of the TeachQuiz metric. However, the reviewers raised several valid concerns. Primary among these was the framework's strict domain limitation to 2D symbolic disciplines (e.g., math and computer science), questioning its generalizability to subjects requiring real-world footage. Additionally, reviewers questioned the robustness of the TeachQuiz metric's unlearning phase and expressed concerns over the computational overhead of the iterative agentic loop.

The authors successfully addressed the empirical concerns by expanding their human evaluation study to 60 participants (showing a strong correlation with TeachQuiz), proving the metric's robustness across multiple frontier VLMs and prompt templates, and demonstrating that Code2Video is more efficient per minute of generated video than purely pixel-based baselines. This work establishes a valuable foundation that the community can meaningfully build upon.